# Massless chiral fields in six dimensions

**Thomas Basile**

Service de Physique de l'Univers, Champs et Gravitation,
Université de Mons, 20 place du Parc, 7000 Mons, Belgium

thomas.basile@umons.ac.be

## Abstract

Massless chiral fields of arbitrary spin in six spacetime dimensions, also known as higher spin singletons, admit a simple formulation in terms of $SU^*(4) \cong SL(2, \mathbb{H})$ tensors. We show that, paralleling the four-dimensional case, these fields can be described using a 0-form and a gauge 2-form, taking values in totally symmetric tensors of $SU^*(4)$. We then exhibit an example of interacting theory that couples a tower of singletons of all integer spin to a background of $\mathfrak{g}$-valued higher spin fields, for $\mathfrak{g}$ an arbitrary Lie algebra equipped with an invariant symmetric bilinear form. Finally, we discuss the formulation of these models in arbitrary even dimensions, as well as their partially-massless counterpart.

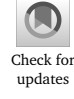
---

# 1 Introduction

Massless fields of arbitrary spin in flat spacetime can be described in plenty of manners, meaning with various sets of field variables, at the free level. Introducing consistent interactions between them is, however, notoriously challenging, and the degree of difficulty of this task may dependent on the set of field variables chosen for this purpose—see e.g. [1,2] for a broad overview of the problem of constructing theories with massless higher spin fields, and [3] for an introductory review.

Recently, a *chiral* theory of higher spin fields in four dimensions has been proposed, first in the light-cone [4] (building upon earlier works [5,6]), then in a Lorentz covariant form [7,8]. This theory exhibits an number of interesting features: For one thing, it admits a formulation around flat space, a notoriously difficult task to achieve for theories with massless higher spin fields. Second, it exhibits signs of good quantum behaviour, such as the absence of loop corrections and UV divergences in flat space [9–11]. Last, but not least, it shows signs of relation with "formality-type" theorems [12–14], in the following sense: not only does it use the Felder–Feigin–Shoikhet 2-cocycle [15] whose explicit formula is obtained as a byproduct of the Tsygan–Shoikhet formality theorem [16,17] (itself a "spin-off" of Kontsevich's celebrated formality theorem [18]) but it also relies on an $A_\infty$-algebra whose brackets are expressed in terms of graphs and weights computed as integrals over particular configuration spaces, and whose higher Jacobi identities can be proved using Stokes' theorem on such spaces—a feature of the algebraic structure and proofs involved in the aforementioned formality theorems.

Chiral higher spin gravity admits two contractions which are higher spin extensions of self-dual Yang–Mills and self-dual gravity [19–22], and hence displays (strong) ties to twistor theory [23–29]. The latter relates a complexified (and compactified) version of spacetime in four dimensions to another complex manifold referred to as "twistor space" (see [30–37] and references therein). This *twistor correspondence* relies on the fact that both manifolds are homogeneous spaces of $SL(4,\mathbb{C})$, which contains the double cover of the Lorentz group $\widetilde{SO}(1,3) \cong SL(2,\mathbb{C})$. This allows for a simple description of massless fields[1] in terms of symmetric tensor-spinors (e.g. [40,41]), at the basis of chiral higher spin gravity.

A similar isomorphism holds in six dimensions: the (double cover of the) Lorentz group is isomorphic to the special linear group of $2 \times 2$ matrices over the quaternions,

$$\widetilde{SO}(1,5) \cong SL(2,\mathbb{H}),$$

the latter being sometimes presented as the *real form* of the special linear group of $4 \times 4$ matrices, $SU^*(4)$. The fundamental and anti-fundamental representation of $SU^*(4)$, which are of dimension 4, correspond to the two chiral spinors of the Lorentz group. More generally, this isomorphism relates totally symmetric $SU^*(4) \cong SL(2,\mathbb{H})$ tensors (of even ranks) to irreducible representations (irreps) of Lorentz algebra characterised by a three-row Young diagram, all of which being of the same length, thereby making up a rectangle, and required to be self-dual in each column. Such tensors, subject to first order differential equations, can be identified with the self-dual part of the curvature of a gauge field with the symmetry of a two-row rectangular Young diagram. This particular class of mixed-symmetry fields[2] is known as "singletons" [66–68] and corresponds exactly to those massless fields in flat space which are also *conformal* [69,70] (see [71] for a pedagogical review of the peculiar features of singletons).

---

[1]Note that massive fields can also be described in terms of tensor-spinors, see e.g. [33,38,39].

[2]Generally, the term *mixed-symmetry fields* refers to those fields in flat space that fall into a representation of the Poincaré group induced from an irrep of the massless little group labelled by a Young diagram with more than one row and one column (see [42] for a review of the representation theory of the Poincaré group, and [43–65] for the description of massless mixed-symmetry fields, in both flat and (anti-)de Sitter spacetimes).

The fact that these fields are of *mixed-symmetry* type constitutes one of the interesting differences with respect to the four-dimensional case. Finding interacting theories for mixed-symmetry fields is also challenging, owing to the increase of gauge symmetries to preserve (in flat space), and to their reducible character, see e.g. [72–75]. They however play a central role in duality-symmetric formulation of gauge theories (see for instance [76–80] and references therein). More specific to six dimensions is the putative existence of the $(2,0)$ superconformal field theory [81–84] whose spectrum consists of a supermultiplet including a self-dual 2-form (see also [85–92] for recent work on the related $(1,0)$ model), or a singleton of spin 1 in our terminology; and the "exotic" supergravity known as the $(4,0)$ theory [93–95] based on a supermultiplet containing a singleton of spin 2, thought of as a graviton, as well as several singletons of spin 1 (see for instance [96–101] for recent work on this topic).

Having a "twistor-inspired" description of singletons at hand begs the question of whether one can reproduce the success of $4d$ chiral higher spin gravity in $6d$? In this paper, we take a first step towards addressing this question by exhibiting a simple formulation for massless higher spin fields in $6d$ dimensions, in terms of field variables which takes advantage of the isomorphism $\mathfrak{so}(1,5) \cong \mathfrak{sl}(2,\mathbb{H}) \cong \mathfrak{su}^*(4)$, and present a couple of examples of interacting theories involving said massless fields.

This paper is organised as follows: in Section 2 we start by reviewing the free description of higher spin singletons in $6d$ and show that they admit a formulation based on a pair of 0-form and a 2-form valued in symmetric $\mathfrak{su}^*(4)$-tensors, then use this formulation as a starting point to couple higher spin singletons to a connection 1-form, and define an higher spin extension thereof. In Section 3, we discuss how to extend the free formulation discussed in $6d$ to arbitrary even dimensions, and argue that the interacting theories also admit a higher-dimensional counterpart. We conclude this paper in Section 4 by discussing various future directions, and propose a "partially-massless" version of the theory discussed in the bulk of this note in Appendix A.

**Conventions.**   In order to lighten expressions containing tensors of arbitrary ranks, we use a notational shorthand common in the higher spin literature which consists in denoting all indices to be symmetrised by the same letter, and indicating the number of such indices in parenthesis if necessary. For instance, we write the components of a symmetric tensor $T$ of rank $n$ as

$$T_{a(n)} = \tfrac{1}{n!} \sum_{\sigma \in \mathcal{S}_n} T_{a_{\sigma_1}...a_{\sigma_n}},$$

where $\mathcal{S}_n$ is the group of permutations of $n$ object. We also write all tensors with the symmetry a given Young diagram in the *symmetric basis*. In plain words, when saying that a tensor has the symmetry of a diagram with $k$ rows of length $\ell_1 \geq \ell_2 \geq \cdots \geq \ell_k > 0$, we mean that this tensor has $k$ groups of symmetric indices, the first one with $\ell_1$ indices, the second with $\ell_2$, etc., and that the symmetrisation of all indices in the $i$th group with one index of the $j$th group with $j > i$ vanishes identically. Such a tensor is therefore denoted by

$$T_{a_1(\ell_1),...,a_k(\ell_k)}, \qquad \text{with} \qquad T_{...,a_i(\ell_i),...,a_i\, a_j(\ell_j-1),...} = 0,$$

using the previously introduced notation. In particular, different groups of indices separated by a comma obey the over-symmetrisation condition given above, while groups of indices separated by a vertical bar do not obey any symmetrisation condition (i.e. $T_{a(\ell)|b}$ *does not vanish* when symmetrising all indices, while $T_{a(\ell),b}$ does). In particular, this means that we denote a pair of antisymmetric indices by separating them with a comma, $T_{a,b} = -T_{b,a}$.

# 2 Six dimensions

## 2.1 Free higher spin singletons

As the chiral formulation of massless higher spin field in $4d$ presented in e.g. [20] relies on the isomorphism of Lie algebras $\mathfrak{so}(1,3) \cong \mathfrak{sl}(2,\mathbb{C})$, its $6d$ counterpart that we will present hereafter is based on the isomorphism $\mathfrak{so}(1,5) \cong \mathfrak{sl}(2,\mathbb{H})$, the latter algebra being also isomorphic to $\mathfrak{su}^*(4)$. Since finite-dimensional representations of both algebras can be conveniently denoted by Young diagrams, we will use the convention that diagrams corresponding to $\mathfrak{so}(1,5)$ irreps will be drawn in blue, while diagrams corresponding to $\mathfrak{su}^*(4) \cong \mathfrak{sl}(2,\mathbb{H})$-irreps will be drawn in orange. For instance, we will write the isomorphism between the vector irrep of $\mathfrak{so}(1,5)$ with the antisymmetric rank 2 tensor of $\mathfrak{su}^*(4)$ as

$$\underset{\mathfrak{so}(1,5)}{\square} \cong \underset{\mathfrak{su}^*(4)}{\square\hspace{-0.65em}\square} \,, \tag{1}$$

and will forget about the subscripts $\mathfrak{so}(1,5)$ and $\mathfrak{su}^*(4)$ from now on.

The above isomorphism tells us that, similarly to the $d = 4$ case, we can use coordinates carrying pairs of antisymmetric spinor indices $A, B, \dots = 1, \dots, 4$, i.e.

$$x^\mu \rightsquigarrow x^{A,B} = -x^{B,A} \,, \tag{2}$$

and the same holds for vector fields and differential forms. Indices can only be raised and lowered by pairs, using the Levi–Civita symbol $\varepsilon_{ABCD}$ of $\mathfrak{su}^*(4)$, via

$$v_{A,B} := \tfrac{1}{2}\,\varepsilon_{ABCD}\,v^{C,D} \qquad \Longleftrightarrow \qquad v^{A,B} = \tfrac{1}{2}\,\varepsilon^{ABCD}\,v_{C,D}\,. \tag{3}$$

This invariant tensor also allows us to reduce any diagram to one with *at most two rows*. Indeed, we can convert any column with three boxes into a single box, via

$$\square\hspace{-0.65em}\square\hspace{-0.65em}\square \ni v_{A,B,C} \quad \longmapsto \quad \tilde{v}^A \propto \varepsilon^{ABCD}\,v_{B,C,D} \in \blacksquare\,, \tag{4}$$

where we filled in the box to keep in mind that the corresponding tensor has an index up: it belongs to the $\mathfrak{su}^*(4)$-irrep *conjugate* to the one denoted by a box. We will adopt this notation in the reminder of this note, namely uncoloured boxes will correspond to tensors of the vector representation, and coloured boxes to tensors of its conjugate. For the antisymmetric Lorentz tensor of rank 2, the relevant isomorphism is

$$\square\hspace{-0.65em}\square \cong \square\hspace{-0.65em}\square\hspace{-0.65em}\square \cong \blacksquare\hspace{-0.65em}\square\,, \tag{5}$$

where in the last step, we have dualised the first column as in (4). Note that the corresponding tensor is traceless, in the sense that the contraction between the upper and lower indices vanishes. One can recognize the adjoint representation of $\mathfrak{so}(1,5)$ and $\mathfrak{su}^*(4)$ in the above isomorphism: one way to present the Lorentz algebra is as being generated by $M_{a,b} = -M_{b,a}$ with $a, b = 0, 1, \dots, 5$ subject to the relations

$$[M_{a,b}, M_{c,d}] = \eta_{bc}\,M_{ad} - \eta_{bd}\,M_{a,c} - \eta_{ac}\,M_{b,d} + \eta_{ad}\,M_{b,c}\,, \tag{6}$$

whereas for $\mathfrak{su}^*(4)$, one can use generators $M^A{}_B$ with $A, B = 1, \dots, 4$ such that $\delta^B_A\,M^A{}_B = 0$, and subject to the relations

$$[M^A{}_B, M^C{}_D] = \delta^C_B\,M^A{}_D - \delta^A_D\,M^C{}_B\,. \tag{7}$$

Recall that the finite-dimensional *irreducible* representations of special linear algebras correspond to tensors whose components may have both upper and lower indices, both groups having the symmetry of a given (generally different) Young diagram, and which are *traceless* in that the contraction of any upper index with any lower one vanishes identically. From now on, we will denote these irreps by drawing the two diagrams, one with its boxes filled in and the other not, and joining the upper right corner of the former one with the lower left corner of the latter—as we did for the adjoint representation (5).

Finally, let us note that for $\mathfrak{so}(1,5)$, totally antisymmetric tensors of rank 3 can be either self-dual or anti-self-dual, which are isomorphic to two different irreps of $\mathfrak{su}^*(4)$, namely

$$
\boxed{}_+ \quad \cong \quad \square\square \,, \qquad \text{and} \qquad \boxed{}_- \quad \cong \quad \square\square\square\square \quad \cong \quad \blacksquare\blacksquare \,, \tag{8}
$$

where in the second term we again dualised the two columns of height 3 to get a symmetric tensor of rank 2.

Assume that the six-dimensional manifold $M$ on which we are working is equipped with a vielbein $e^{A,B}$, which is a 1-form taking values in the antisymmetric irrep of $\mathfrak{su}^*(4)$. Following (3), the vielbein with all indices lowered is defined as,

$$
e_{A,B} := \tfrac{1}{2}\, \varepsilon_{ABCD}\, e^{C,D} \,, \tag{9}
$$

and we can use it to define a basis for the space of 2-forms, via

$$
\Sigma_A{}^B := e_{A,C} \wedge e^{C,B} \equiv \tfrac{1}{2}\, \varepsilon_{ACDE}\, e^{B,C} \wedge e^{D,E} \,. \tag{10}
$$

In particular, one can easily see that it is traceless, so that these 2-forms do carry the irrep $\square$, and a direct computation leads to

$$
e^{A,B} \wedge e^{C,D} = \tfrac{1}{2}\left( \varepsilon^{\times AB[C}\Sigma_\times{}^{D]} - \varepsilon^{\times CD[A}\Sigma_\times{}^{B]} \right) \qquad \Longleftrightarrow \qquad e^{A,B} \wedge e_{C,D} = 2\,\delta^{[A}_{[C}\Sigma_{D]}{}^{B]} \,, \tag{11}
$$

thereby allowing us to write any product of 1-forms in this basis. We can proceed further and define the following anti/self-dual 3-forms,

$$
H^{AA} := e^{A,C} \wedge \Sigma_C{}^A \equiv e_{B,C} \wedge e^{A,B} \wedge e^{A,C} \,, \tag{12a}
$$

$$
H_{AA} := \Sigma_A{}^C \wedge e_{C,A} \equiv e^{B,C} \wedge e_{A,B} \wedge e_{A,C} \,, \tag{12b}
$$

which form a basis of the space of 3-forms. Once again, a direct computation yields

$$
e^{A,B} \wedge \Sigma_D{}^C = \tfrac{1}{3}\, \varepsilon^{ABC\times} H_{\times D} - \tfrac{2}{3}\, \delta^{[A}_D H^{B]C} \,, \tag{13}
$$

thereby expressing the product of any 1-form with any other 2-form in the basis of 3-forms defined in (12). Next, we can define a basis of 4-forms by taking products of the previously obtained basis of 1-, 2- and 3- forms, projected on the correct $\mathfrak{su}^*(4)$-irrep (that is $\square$), namely

$$
H_{AC} \wedge e^{C,B} = e_{A,C} \wedge H^{CB} = \Sigma_A{}^C \wedge \Sigma_C{}^B \,. \tag{14}
$$

Yet again, a direct computation gives us

$$
e^{A,B} \wedge H_{CC} = 2\, \Sigma_C{}^D \wedge \Sigma_D{}^{[A}\delta^{B]}_C \,, \qquad \text{and} \qquad e_{A,B} \wedge H_{CC} = \Sigma_C{}^D \wedge \Sigma_D{}^\times \varepsilon_{\times ABC} \,, \tag{15}
$$

so that, combined with the previous results, we can decompose products of forms up to degree 4 in the basis introduced so far. In particular, one can check that

$$
H_{AA} \wedge e_{A,B} = 0 \qquad \Longrightarrow \qquad H_{AA} \wedge \Sigma_A{}^B = 0 \,, \tag{16}
$$

as a consequence of (11). Let us conclude this discussion on forms by pointing out that the self-dual 3-forms are "orthogonal" to one another, whereas the product of self-dual and anti-self-dual 3-forms is proportional to a top form on $M$,

$$
H_{AA} \wedge H_{BB} = 0 \,, \qquad H_{AA} \wedge H^{BB} \propto \delta^B_A \delta^B_A \, \mathrm{vol}_M \,, \tag{17}
$$

where $\mathrm{vol}_M$ denotes the volume form induced by the vielbein.

**Massless equations in six dimensions.** Around flat spacetime in $d = 6$ dimensions, there are several types of massless fields one can consider, since their spin is determined by an irrep of $Spin(4) \cong SU(2) \times SU(2)$, and hence labelled by *two (half-)integers*. This opens the possibility of considering *mixed-symmetry* fields, which are fields whose little group irrep corresponds to a two-row Young diagram of $Spin(4)$, or equivalently, for which the two $SU(2)$ irreps are different. The "limiting case" when exactly one of this $SU(2)$ irreps is trivial, corresponding to *rectangular* two-row Young diagrams.

Such fields can be realised by Lorentz tensors, with the symmetry of the same two-row diagram, and enjoying some gauge symmetries. The simplest example is probably a massless 2-form $B$, whose curvature 3-form is denoted $H$,

$$B = B_{\mu,\nu}\, \mathrm{d}x^{\mu} \wedge \mathrm{d}x^{\nu} \in \Omega_M^2 \cong \boxed{\phantom{x}}, \qquad H = \mathrm{d}B \in \Omega_M^3 \cong \boxed{\phantom{x}}, \tag{18}$$

on which we can impose some self-duality or anti-self-duality condition. More generally, one can consider the spin $s$ version, meaning a gauge field $\varphi$ with the symmetry of a rectangular two-row diagram of length $s$,

$$\varphi_{\mu(s),\nu(s)} \longleftrightarrow \boxed{\phantom{s}} \cong \boxed{\phantom{s}} \cong \boxed{\phantom{s}} \longleftrightarrow \varphi_{A(s)}{}^{B(s)}, \tag{19}$$

whose curvature is a rectangular three-row diagram,

$$F_{\mu(s),\nu(s),\rho(s)} = \underbrace{\partial_\rho \ldots \partial_\rho}_{s} \varphi_{\mu(s),\nu(s)} + \cdots \longleftrightarrow \boxed{\phantom{s}}$$

$$\cong \boxed{\phantom{2s}} \longleftrightarrow F_{A(2s)} = \underbrace{\partial_{A,B} \ldots \partial_{A,B}}_{s} \varphi_{A(s)}{}^{B(s)}, \tag{20}$$

$$\oplus \boxed{\phantom{2s}} \longleftrightarrow F^{B(2s)} = \underbrace{\partial^{A,B} \ldots \partial^{A,B}}_{s} \varphi_{A(s)}{}^{B(s)},$$

on which we can also impose anti/self-duality conditions. In terms of $\mathfrak{su}^*(4)$, this becomes quite simple as it amounts to keeping either $F_{A(2s)}$ or $F^{A(2s)}$ only.

The aforementioned massless fields are known to be conformal, and are sometimes referred to as spin $s$ singletons [66, 68].[3] The equations of motion for a singleton of spin $s$ and say positive chirality, in terms of $\mathfrak{su}^*(4)$ tensors read [102, 111, 112][4]

$$\partial_{B,C} \Psi^{A(2s-1)C} \approx 0, \tag{21}$$

or in terms of a potential,[5]

$$\partial_{B,A} \Phi_{A(2s-1)}{}^B \approx 0, \tag{22}$$

where $\Phi_{A(2s-1)}{}^B$ is traceless, and we used the symbol "$\approx$" merely as a way to highlight the fact that solutions of these equations are *on-shell* fields. The last equation is invariant under the gauge symmetry

$$\delta_\xi \Phi_{A(2s-1)}{}^B = \partial^{B,C} \xi_{A(2s-1),C} - \tfrac{2s-1}{2s+2} \delta_A^B \partial^{C,D} \xi_{A(2s-2)C,D}, \tag{23}$$

---

[3]Note that they were already identified in [69], and discussed in more details recently [71, 102, 103]. See also [103–108] for more details concerning their symmetry algebra, and [109, 110] for reviews of higher spin algebras in general dimensions.

[4]See also [103] for the formulation of higher spin singletons in *arbitrary even* dimensions using Howe duality.

[5]The two are typically related by $\Psi^{A(2s)} = (\partial^{A,B})^{2s-1} \Phi_{B(2s-1)}{}^A$ in flat space, with $\Psi^{A(2s)}$ corresponding to a chiral half of the field strength in (20).

where $\xi_{A(2s-1),B}$ is of "hook-type" symmetry,  (one should use the identity $\partial_{A,B}\partial^{B,C} \propto \delta_A^C \Box$ and the symmetry property of $\xi$ to verify this statement). This gauge symmetry is reducible, as one can check that gauge parameters of the form

$$\mathring{\xi}_{A(2s-1),B} := \partial_{B,A}\zeta_{A(2s-2)},$$ (24)

lead to trivial gauge transformations, i.e. $\delta_{\mathring{\xi}}\Phi_{A(2s-1)}{}^B = 0$. These wave equations, (21) and (22), can be derived from the first order action

$$S[\Psi, \Phi] = \int_M \mathrm{vol}_M \ \Psi^{A(2s)} \partial_{B,A}\Phi_{A(2s-1)}{}^B,$$ (25)

where $\mathrm{vol}_M$ denotes the volume form on $d = 6$ Minkowski spacetime.

Let us point out that, contrary to the $4d$ case, the fields $\Psi$ and $\Phi$ both describe a singleton of spin $s$ with the *same chirality* (we will comment further on this point in Section 3), and hence the above functional and its reformulation that will be presented below are not actions for self-dual 2-forms when $s = 1$. Actions for self-dual $p$-forms have been studied by many authors, see for instance [113–125] and references therein.

**Action principle for singletons.** We can re-write a little bit the previous action by embedding the field $\Phi_{A(2s-1)}{}^B$ in a 2-form, via

$$\varpi_{A(2s-2)} = \Sigma_C{}^B \Phi_{A(2s-2)B}{}^C,$$ (26)

and assuming that we are now working on a constant curvature background equipped with a torsion-free connection, i.e.[6]

$$\nabla^2 = \Sigma_A{}^B \rho(M^A{}_B),$$ (27)

where $M^A{}_B$ are the $\mathfrak{su}^*(4)$-generators and $\rho$ the representation of this algebra on the section of the bundle acted upon by the curvature. Concretely, it verifies

$$\begin{aligned}
\nabla^2 \varphi_{A(k),B(l)}{}^{C(m),D(n)} = &-k \Sigma_A{}^\times \varphi_{A(k-1)\times,B(l)}{}^{C(m),D(n)} - l \Sigma_B{}^\times \varphi_{A(k),B(l-1)\times}{}^{C(m),D(n)} \\
&+ m \Sigma_\times{}^C \varphi_{A(k),B(l)}{}^{C(m-1)\times,D(n)} + n \Sigma_\times{}^D \varphi_{A(k),B(l)}{}^{C(m),D(n-1)\times},
\end{aligned}$$ (28)

when acting on a generic tensor $\varphi$ which is "hook-symmetric" in both its upper and lower indices. The covariant differential of the 2-form $\varpi$, which is a 3-form and hence contain both a self-dual and an anti-self-dual part, reads

$$\nabla\varpi_{A(2s-2)} = \tfrac{2}{3}\nabla^{B,C}\Phi_{A(2s-2)B}{}^C H_{CC} + \tfrac{2}{3}H^{BB}\nabla_{B,C}\Phi_{A(2s-2)B}{}^C,$$ (29)

as can be checked using (11). Using the identity (17), we can get rid of the first term and keep only the second one in previous equation by multiplying it by the self-dual 3-form,

$$H_{AA} \wedge \nabla\varpi_{A(2s-2)} \propto \mathrm{vol}_M \nabla_{B,A}\Phi_{A(2s-1)}{}^B,$$ (30)

thereby reproducing the part of the integrand of the action (25) to be contracted with $\Psi$.

Having re-expressed the action for a spin-$s$ singleton in terms of differential forms, by embedding $\Phi_{A(2s-1)}{}^B$ in the 2-form $\varpi_{A(2s-2)}$, we can now promote the latter to an independent field—that we shall denote by $\omega_{A(2s-2)}$ so as to avoid any confusion—and postulate the free action

$$S[\Psi, \omega] = \int \Psi^{A(2s)} H_{AA} \wedge \nabla\omega_{A(2s-2)},$$ (31)

---

[6]We implicitly normalised this curvature for simplicity.

which looks exactly like the one proposed in [20], upon replacing $SL(2,\mathbb{C})$ spinor indices for $SU^*(4) \cong SL(2,\mathbb{H})$ ones. This action is invariant under the gauge symmetries

$$\delta_{\xi,\eta}\omega_{A(2s-2)} = \nabla\xi_{A(2s-2)} + e_{A,B}\,\eta_{A(2s-3)}{}^B\,, \tag{32}$$

as can be verified using the identities (16). Note that, as before, these gauge transformations are reducible: gauge parameters of the form[7]

$$\mathring{\xi}_{A(2s-2)} = \nabla\zeta_{A(2s-2)}\,, \qquad \text{and} \qquad \mathring{\eta}_{A(2s-3)}{}^B = (2s-2)\,e^{B,C}\,\zeta_{A(2s-3)C}\,, \tag{33}$$

leave $\omega$ invariant, $\delta_{\mathring{\xi},\mathring{\eta}}\omega = 0$, for any 0-form $\zeta_{A(2s-2)}$. These gauge symmetries allows one to gauge away the unwanted components contained in $\omega_{A(2s-2)} = \Sigma_C{}^B\,\omega_{A(2s-2)|B}{}^C$. To be more precise, the decomposition into irreducible components of this 2-form reads

$$\boxed{\phantom{x}}\otimes\boxed{2s-2}\;\cong\;\boxed{2s-1}\;\oplus\;\boxed{2s-2}\;\oplus\;\boxed{2s-2}\;\oplus\;\boxed{2s-3}\,, \tag{34}$$

or explicitly

$$\omega_{A(2s-2)} = \Sigma_C{}^B\left(\Phi_{A(2s-2)B}{}^C + \Theta_{A(2s-2),B}{}^C\right) + \Sigma_A{}^B\left(\zeta_{A(2s-3)B} + \Xi_{A(2s-3),B}\right), \tag{35}$$

where $\Phi$ is the gauge field we would like to keep, and $\Theta$, $\zeta$ and $\Xi$ are 0-forms parameterising the aforementioned unwanted components. As it turns out, the latter can be gauged away thanks to the algebraic/shift symmetry generated by $\eta$. Indeed, decomposing this gauge parameter into its irreducible components as well, one finds that it contains the irreps

$$\boxed{\phantom{x}}\otimes\boxed{2s-3}\;\cong\;\boxed{2s-2}\;\oplus\;\boxed{2s-4}\;\oplus\;\boxed{2s-2}\;\oplus\;\boxed{2s-3}\,, \tag{36}$$

or explicitly

$$\begin{aligned}\eta_{A(2s-3)}{}^B = e^{C,D}\Big(&\Theta_{A(2s-3)C,D}{}^B + \varepsilon_{\times CDA}\Lambda_{A(2s-4)}{}^{B\times} + \delta_C^B\,\zeta_{A(2s-3)D}\\ &+ \delta_C^B\,\Xi_{A(2s-3),D} - \tfrac{2s-3}{2s}\,\delta_A^B\Xi_{A(2s-4)C,D}\Big), \end{aligned} \tag{37}$$

where $\Theta$, $\zeta$ and $\Xi$ are *not* the same tensors as those appearing in the decomposition of $\omega$, but carry the same irrep (which is why we choose to denote them by the same symbols). They can therefore be used to gauge away the unwanted components in $\omega$. The last tensor $\Lambda$ fortunately *does not appear* in the gauge transformation (32), as can be checked using the identity (11). Finally, the differential gauge parameter $\xi$ can be decomposed into,

$$\boxed{\phantom{x}}\otimes\boxed{2s-2}\;\cong\;\boxed{2s-1}\;\oplus\;\boxed{2s-3}\,, \tag{38}$$

i.e. it is parameterised by two tensors via

$$\xi_{A(2s-2)} = e^{B,C}\,\xi_{A(2s-2)B,C} + e_{B,A}\,\lambda_{A(2s-3)}{}^B\,, \tag{39}$$

where we abused notation by using the same symbol for the component of $\xi_{A(2s-2)}$ which corresponds to the gauge parameter involved in the gauge transformation (23) of $\Phi_{A(2s-1)}{}^B$, that is $\xi_{A(2s-1),B}$. One can check, by direct computation, that the gauge transformation generated by the extra component $\lambda$ *does not* affect the $\Phi$ component of $\omega$, but only the pure gauge components $\Theta$, $\zeta$ and $\Xi$.

---

[7]Note that in flat space, the reducibility parameter for $\eta$ becomes trivial, i.e. $\mathring{\eta} = 0$. Indeed, the role of $\mathring{\eta}$ is to compensate for the curvature term appearing due to $\mathring{\xi}$. This limit is not explicit here due our choice of absorbing the cosmological constant in the definition of $H_A{}^B$.

The equations of motion resulting from the previous action are given by

$$0 \approx \nabla \Psi^{A(2s-2)BB} \wedge H_{BB}, \tag{40a}$$

$$0 \approx \nabla \omega_{A(2s-2)} \wedge H_{AA}, \tag{40b}$$

which are equivalent to the equations (21) and (22). Let us start with the first one, which reads

$$0 \approx \nabla_{C,D} \Psi^{A(2s-2)BB} e^{C,D} \wedge H_{BB} = 2 \Sigma_B{}^E \wedge \Sigma_E{}^C \nabla_{C,D} \Psi^{A(2s-2)BD}, \tag{41}$$

upon using (15), and hence does reproduce (21). We already saw that the second equation is nothing but the equation for $\Phi$ embedded in $\omega$, see (30), and that all other components can be gauged away, hence the pair of wave equation (40) is equivalent to the equations (21) and (22).

**Reducibility of the gauge symmetries.** Before going further, let us spend more time on the free gauge symmetries (32), with a special attention to their *reducible* character. As discussed above, this last property is necessary in order to ensure that the free action (31) and the wave equations that stem from it describes the propagation of the correct number of degrees of freedom, namely $2s + 1$ per fields (i.e. for $\Psi$ and for $\omega$). This corresponds to the dimension of the irrep of the $6d$ massless little group $Spin(4)$, with highest weight $(s, \pm s)$, and can be checked using the algorithm spelled out in [126].[8]

We should start by remarking that the free description outlined in the previous section can be understood in terms of the Lie algebra $\mathfrak{so}(2,5)$, and its finite-dimensional representation labelled by the three-row Young diagram of length $s - 1$, whose decomposition under the Lorentz subalgebra reads

$$\begin{array}{c}\boxed{\begin{array}{c}s-1\\ \hline \\ \hline\end{array}}\end{array}\begin{array}{c}\mathfrak{so}(2,5)\\ \downarrow\\ \mathfrak{so}(1,5)\end{array}\bigoplus_{\sigma=1}^{s}\boxed{\begin{array}{cc}s-1\\ \hline s-\sigma\end{array}}_{+}\oplus\boxed{\begin{array}{cc}s-1\\ \hline s-\sigma\end{array}}_{-}\cong\bigoplus_{\sigma=0}^{2s-2}\boxed{\sigma}\boxed{2s-2-\sigma}, \tag{42}$$

where it should be understood that the two-row Young (sub)diagram of $\mathfrak{so}(1,5)$, with both rows of length $s - 1$, appears only once. To do so, one can start by parameterising the space of solutions of the equations of motion for the 2-form $\omega$, as

$$H_{AA} \wedge \nabla \omega_{A(2s-2)} \approx 0 \qquad \Longrightarrow \qquad \nabla \omega_{A(2s-2)} + e_{A,B} \wedge \omega_{A(2s-3)}{}^B = H^{BB} C_{A(2s-2)BB}, \tag{43}$$

where $\omega_{A(2s-3)}{}^B$ is a 2-form and $C_{A(2s)}$ a 0-form. Having introduced these two new fields, which describe the *unconstrained* part of the first derivatives of $\omega_{A(2s-2)}$, one should try and characterise them. To do so, we can differentiate the above equation and parameterise the first derivatives of the new fields in a way consistent with the previous equation. For the 2-form $\omega_{A(2s-3)}{}^B$, this leads to

$$\nabla \omega_{A(2s-3)}{}^B + e_{A,C} \wedge \omega_{A(2s-4)}{}^{BC} + (2s-2) e^{B,C} \wedge \omega_{A(2s-3)C} = 0, \tag{44}$$

where we have introduced a new 2-form $\omega_{A(2s-4)}{}^{BB}$. Its appearance is due to the freedom afforded by the fact that the operation which consists in wedging a form $\alpha_{A(2s-3)}{}^B$ with a vielbein and contracting one of its index with $B$ and symmetrising the other with the $A$'s, i.e. the operation $\alpha_{A(2s-3)}{}^B \longmapsto e_{A,B} \wedge \alpha_{A(2s-3)}{}^B$, has a non-trivial kernel. The 3-form $e_{A,C} \wedge \omega_{A(2s-4)}{}^{BC}$ belongs to this kernel, and hence is consistent with the equation (43) for $\nabla \omega_{A(2s-2)}$ in that it does not appear when differentiating it.

---

[8]Let us also mention that the fact that both $\Psi$ and $\Phi$ (and hence by extensions $\omega$) propagate the same number of degrees of freedom follows from the general analysis of [127]. I am thankful to an anonymous referee for pointing out these references, as well as insightful remarks about the counting of degrees of freedom.

The spectrum of $\mathfrak{so}(1,5) \cong \mathfrak{su}^*(4)$-valued 2-forms appearing in this process is exactly that of the branching rule (42) above. Moreover, these 2-forms *do assemble* into the $\mathfrak{so}(2,5)$-irrep labelled by the three-row diagram of length $s-1$, as can be verified by analysing the terms proportional to the vielbein $e_{A,B}$ in their equations of motion. Indeed, it turns out that the latter stem from the action of the "transvection" generators, i.e. the generators of the complement of the Lorentz subalgebra in $\mathfrak{so}(2,5)$. To see that, it is useful to introduce auxiliary variables $y^A$ and $\bar{y}_A$ to contract the indices carried by the 2-forms with, i.e. we define

$$\omega^{(-t)} := \frac{1}{(2s-t-2)!\, t!}\, y^{A(2s-t-2)} \bar{y}_{B(t)}\, \omega_{A(2s-t-2)}{}^{B(t)}\,, \tag{45}$$

for $t = 0, 1, \dots, 2s-2$. The action of $\mathfrak{su}^*(4)$ can be represented by

$$\rho\left(M^A{}_B\right) = y^A \frac{\partial}{\partial y^B} - \bar{y}_B \frac{\partial}{\partial \bar{y}_A}\,, \tag{46}$$

while the transvection generators by

$$\rho\left(P^{A,B}\right) = y^{[A}\, \bar{\partial}^{B]} + \tfrac{1}{2}\, \varepsilon^{ABCD}\, \bar{y}_C\, \partial_D\,. \tag{47}$$

Upon introducing the operators

$$\sigma_+ := e_{A,B}\, y^A\, \bar{\partial}^B\,, \qquad \sigma_- := e^{A,B}\, \bar{y}_A\, \partial_B\,, \qquad \sigma := \sigma_+ + \sigma_- \equiv e_{A,B}\, \rho\left(P^{A,B}\right)\,, \tag{48}$$

one can rewrite the equations of motion for the 2-forms $\omega^{(-t)}$ as

$$\left(\nabla + \sigma\right)\omega^{(-t)} = \delta_{t,0}\, H^{BB}\, C_{A(2s-2)BB}\, \frac{1}{(2s-2)!}\, y^{A(2s-2)}\,. \tag{49}$$

The operators $\sigma_\pm$ satisfy

$$(\sigma_\pm)^2 = 0\,, \qquad \{\sigma_+, \sigma_-\} = \Sigma_A{}^B\, \rho\left(M^A{}_B\right) \qquad \Longrightarrow \qquad \left(\nabla + \sigma\right)^2 = 0\,, \tag{50}$$

and relate 2-forms $\omega^{(k\mp1)}$ to $\omega^{(k)}$, i.e. they increase/decrease the upper index by one unit. Since the equations of motion are given in terms of the action of a differential, $\nabla + \sigma$, on the 2-forms, their gauge symmetry simply takes the form of exact terms,

$$\delta_\xi \omega^{(-t)} = \nabla \xi^{(-t)} + \sigma_- \xi^{(-t+1)} + \sigma_+ \xi^{(-t-1)}\,, \tag{51}$$

where $\xi^{(-k)}$ is a 1-form taking values in the same irrep as $\omega^{(-k)}$—a diagram with $k$ coloured boxes. In particular, for $t = 0$, two parameters enter the gauge transformation of $\omega^{(0)}$ namely $\xi^{(0)}$ and $\xi^{(-1)}$, which correspond to $\xi$ and $\eta$ as in the expression (32) given above. Similarly, the space of reducible gauge parameters is simply that of exact ones, i.e. parameters of the form

$$\mathring{\xi}^{(-t)} = \nabla \zeta^{(-t)} + \sigma_- \zeta^{(-t+1)} + \sigma_+ \zeta^{(-t-1)}\,, \tag{52}$$

where $\zeta^{(-k)}$ are now 0-forms taking values in the same irrep as $\xi^{(-k)}$. For $t = 0$, we find that two parameters $\zeta^{(0)}$ and $\zeta^{(-1)}$ enter the reducibility of $\xi^{(0)}$, while for $t = -1$, these two parameters as well as a third one, $\zeta^{(-2)}$, are involved in the reducibility of $\xi^{(-1)}$. In this case, $\zeta^{(0)}$ corresponds to the reducibility parameter $\zeta$ used for the reduction (33) gauge symmetry generated by $\xi$ and $\eta$, and constitute a consistent truncation of the above general formula (51). The various fields, gauge and reducibility parameters discussed are summarised below,

together with (some of) the operators relating them.

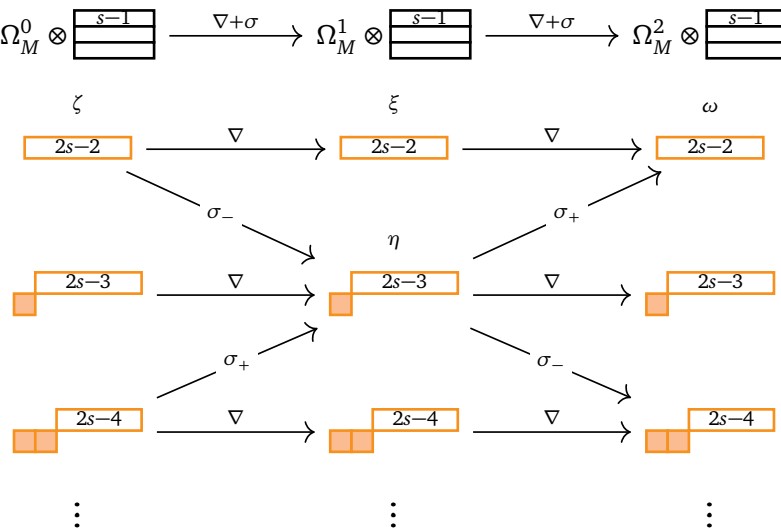

Note that we did not draw all diagonal arrows in the above picture, for the sake of clarity.

Splitting the action of the transvection generators into two operators, $\sigma_{\pm}$, is a standard and quite efficient way of analysing the content of equations of motion which amount to fixing the covariant derivative of a $p$-form valued in a finite-dimensional module of some (isometry) Lie algebra (to be non-vanishing only in a specific subspace). In particular, one finds that the cohomology of the $\sigma_-$ operator allows one to identify more precisely this field content (see e.g. [128,129], and references therein).

## 2.2 A word about interactions

Now let us proceed with constructing a couple of examples of interacting deformation of the previous free action. As a first example, we will try to leverage the resemblance of this free description with the $4d$ treatment of massless higher spin fields based on $\mathfrak{so}(1,3) \cong \mathfrak{sl}(2,\mathbb{C})$ and construct an action which maybe thought as the $6d$ counterpart of the higher spin extension of self-dual Yang–Mills introduced in [19,20].

**Warming-up: coupling to a connection 1-form.** First, let us show that the higher spin singletons discussed in the previous section can be coupled to a spin-1 gauge field, that is a $\mathfrak{g}$-valued gauge field $A \in \Omega_M^1 \otimes \mathfrak{g}$. The Lie algebra $\mathfrak{g}$ (whose indices and generators we have suppressed for clarity) is assumed, as usual, to be equipped with an ad-invariant symmetric bilinear form denoted by $\langle -,- \rangle$. In other words, this bilinear form satisfies

$$\langle x, y \rangle = \langle y, x \rangle, \qquad \text{and} \qquad \langle [x,y]_{\mathfrak{g}}, z \rangle = \langle x, [y,z]_{\mathfrak{g}} \rangle, \tag{53}$$

for any elements $x, y, z \in \mathfrak{g}$. Promoting the field $\Psi$ and $\omega$ to be $\mathfrak{g}$-valued, and defining the covariant derivative and associated field strength,

$$D := \nabla + [A,-]_{\mathfrak{g}}, \qquad F := \mathrm{d}A + \tfrac{1}{2}[A,A]_{\mathfrak{g}}, \tag{54}$$

allows us to write the following action,

$$S_{\text{YM–like}}[\Psi, \omega; A] = \int_M \langle \Psi^{A(2s)}, H_{AA} \wedge D\omega_{A(2s-2)} \rangle, \tag{55}$$

which is clearly invariant under the usual gauge transformations of a connection 1-form,

$$\delta_\epsilon A = D\epsilon\,, \qquad \delta_\epsilon \omega_{A(2s-2)} = [\omega_{A(2s-2)}, \epsilon]_{\mathfrak{g}}\,, \qquad \delta_\epsilon \Psi^{A(2s)} = [\Psi^{A(2s)}, \epsilon]_{\mathfrak{g}}\,, \tag{56}$$

with $\epsilon \in \Omega_M^0 \otimes \mathfrak{g}$. On top of that, it can be made invariant under

$$\delta_{\xi,\eta} \omega_{A(2s-2)} = D\xi_{A(2s-2)} + e_{A,B} \wedge \eta_{A(2s-3)}{}^B\,, \tag{57}$$

where both $\xi$ and $\eta$ are $\mathfrak{g}$-valued 1-forms, by adding a BF term

$$S_{\text{BF}}[A,B] = \int_M \langle B, F \rangle\,, \qquad \delta_\epsilon B = [B, \epsilon]_{\mathfrak{g}}\,, \tag{58}$$

where $B \in \Omega_M^4 \otimes \mathfrak{g}$ transform in the adjoint representation of $\mathfrak{g}$. This ensures that this BF term is invariant under the gauge transformations generated by $\epsilon$. More importantly, the introduction of the 4-form $B$ allow us to render the sum of the actions $S_{\text{YM-like}}$ (55) and $S_{\text{BF}}$ (58) gauge-invariant under the transformations (56) generated by the parameters $\xi$ and $\eta$, by a suitable choice of transformation for $B$. More precisely, one can check that the variation of $S_{\text{YM-like}}[\Psi, \omega; A]$ under the transformations generated by $\xi$ and $\eta$ is compensated by that of $S_{\text{BF}}[A, B]$, if one assumes that $A$ is inert and

$$\delta_{\xi,\eta} B = -\big[\Psi^{A(2s)}, \xi_{A(2s-2)}\big]_{\mathfrak{g}} \wedge H_{AA}\,. \tag{59}$$

Doing so simply requires using the identities (16), and the ad-invariance of the symmetric bilinear form (53) on $\mathfrak{g}$.

Finally, let us remark that the gauge transformations of $\omega$ are reducible *on-shell*, as the choice of gauge parameters

$$\mathring{\xi}_{A(2s-2)} = D\zeta_{A(2s-2)}\,, \qquad \mathring{\eta}_{A(2s-3)}{}^B = (2s-2)\, e^{B,C}\, \zeta_{A(2s-3)C}\,, \tag{60}$$

in the presence of the gauge field $A$, leads to trivial gauge transformations, for $F \approx 0$.[9]

**Higher spin extension.** Paralleling the construction of [20] (see also [130] for the partially-massless case), let us introduce the generating functions

$$\Omega_M^0 \otimes \mathbb{C}[\bar{y}]^{\mathbb{Z}_2} \otimes \mathfrak{g} \ni \Psi(x|\bar{y}) := \sum_{s=1}^\infty \frac{1}{(2s)!}\, \bar{y}_{A(2s)}\, \Psi^{A(2s)}(x)\,,$$

$$\Omega_M^2 \otimes \mathbb{C}[y]^{\mathbb{Z}_2} \otimes \mathfrak{g} \ni \omega(x|y) := \sum_{s=1}^\infty \frac{1}{(2s-2)!}\, y^{A(2s-2)}\, \omega_{A(2s-2)}(x)\,, \tag{61}$$

both of which are again assumed to take values in a Lie algebra $\mathfrak{g}$. Let us also define $H := \frac{1}{2} H_{AA}\, y^A y^A$, with the help of which we can write the sum of the previous action (55) for all integer spin $s \geq 1$ in one go as,

$$S[\Psi, \omega; A, B] = \int_M p \circ \langle \Psi, H \wedge D\omega \rangle + \langle B, F \rangle\,, \tag{62}$$

where

$$p : \mathbb{C}[\bar{y}] \otimes \mathbb{C}[y] \longrightarrow \mathbb{C}\,,$$

$$f(\bar{y}) \otimes g(y) \longmapsto \sum_{n=1}^\infty \frac{1}{n!}\, f^{A(n)} g_{A(n)}\,, \tag{63}$$

---

[9]Note that one can even make the gauge transformations reducible *off-shell*, by adding to (57) the term $\frac{1}{3(s-1)} \big[F, e_{A,B}^\mu \eta_{\mu|A(2s-3)}{}^B\big]_{\mathfrak{g}}$, which will imply a similar modification of the gauge transformation of $B$.

is the pairing between totally symmetric $\mathfrak{su}^*(4)$-tensors and their conjugate. The gauge parameters can also be packaged into generating functions

$$\xi = \sum_{s=1}^{\infty} \tfrac{1}{(2s-2)!} \, y^{A(2s-2)} \, \xi_{A(2s-2)} \,, \qquad \eta = \sum_{s=2}^{\infty} \tfrac{1}{(2s-3)!} \, y^{A(2s-3)} \, \bar{y}_B \, \eta_{A(2s-3)}{}^{B} \,, \tag{64}$$

both of which are 1-form and valued in the Lie algebra $\mathfrak{g}$, so that the gauge symmetries of $\omega$ and $\Psi$ can be put into the compact form

$$\delta_{\xi,\eta,\epsilon} \omega = D\xi + \sigma_+ \eta + [\omega, \epsilon]_{\mathfrak{g}} \,, \qquad \delta_\epsilon \Psi = [\Psi, \epsilon]_{\mathfrak{g}} \,, \tag{65}$$

where everywhere $[-,-]_{\mathfrak{g}}$ should be understood as the $\mathbb{C}[y,\bar{y}]$-linear extension of the Lie bracket of $\mathfrak{g}$, and the operator $\sigma_+$ is the operator defined previously—see (48). Similarly, the reducibility parameters for these gauge symmetries can be simply written as

$$\mathring{\xi} = D\zeta \,, \qquad \text{and} \qquad \mathring{\eta} = \sigma_- \zeta \,, \tag{66}$$

where $\zeta \in \Omega_M^0 \otimes \mathbb{C}[y]^{\mathbb{Z}_2} \otimes \mathfrak{g}$. Finally, the gauge transformations of the field $B$ read

$$\delta_{\xi,\epsilon} B = -p\big([\Psi,\xi]_{\mathfrak{g}} \wedge H\big) + [B,\epsilon]_{\mathfrak{g}} \,, \tag{67}$$

where the pairing $p$ allowed us to package the contribution required from all spin $s \geq 1$. This also highlights the fact that neither the gauge field $A$ nor the $B$-field take values in the algebras of (even) polynomials in $y^A$ or $\bar{y}_A$, but this possibility is interesting to explore.

So let us now replace $A$ and $B$ with 1- and 4- forms taking values in $\mathbb{C}[y]^{\mathbb{Z}_2} \otimes \mathfrak{g}$, i.e.

$$\begin{aligned}
\Omega_M^1 \otimes \mathbb{C}[y]^{\mathbb{Z}_2} \otimes \mathfrak{g} \ni \mathcal{A} &:= \sum_{s=1}^{\infty} \tfrac{1}{(2s-2)!} \, y^{A(2s-2)} A_{A(2s-2)} \,, \\
\Omega_M^4 \otimes \mathbb{C}[\bar{y}]^{\mathbb{Z}_2} \otimes \mathfrak{g} \ni \mathcal{B} &:= \sum_{s=1}^{\infty} \tfrac{1}{(2s-2)!} \, \bar{y}_{A(2s-2)} B^{A(2s-2)} \,,
\end{aligned} \tag{68}$$

and denote the covariant derivative and curvature of $\mathcal{A}$ as

$$\mathcal{D} := \nabla + [\mathcal{A}, -]_{\mathfrak{g}} \,, \qquad \mathcal{F} := \nabla \mathcal{A} + \tfrac{1}{2} [\mathcal{A}, \mathcal{A}]_{\mathfrak{g}} \,, \tag{69}$$

so that we can now consider the action

$$S[\Psi, \omega; \mathcal{A}, \mathcal{B}] = \int_M p \circ \big\langle \Psi, H \wedge \mathcal{D}\omega \big\rangle + p \circ \big\langle \mathcal{B}, \mathcal{F} \big\rangle \,. \tag{70}$$

The 1-form fields making up $\mathcal{A}$, though forming a topological background here since they appear as part of a BF term, can describe hook-symmetric $\mathfrak{su}^*(4)$-tensors

$$\varphi_{A(2s-2)B,C} \quad \longleftrightarrow \quad \boxed{\phantom{xx} 2s-1 \phantom{xx}}\,\square \;\cong\; \overline{\underset{+}{\boxed{\begin{matrix} s \\ s-1 \end{matrix}}}} \,. \tag{71}$$

Indeed, such tensors are embedded in the 1-form as

$$A_{A(2s-2)} = e^{B,C} \, \varphi_{A(2s-2)B,C} + (\dots) \,, \tag{72}$$

where the $(\dots)$ denote other irreps that can be gauged away by a shift symmetry (similar to the one considered for $\omega$). If instead of a zero-curvature condition $\mathcal{F} = 0$, one considers the linear equations $H \wedge \nabla \mathcal{A} \approx 0$, the solution space describes the irrep corresponding to a massless mixed-symmetry field labelled by the above Young diagram.

Let us come back to the previous action and postulate the same gauge transformations for $\omega$ and $\mathcal{A}$ as before, except we replace the covariant derivative $D \to \mathcal{D}$, i.e.

$$\delta_{\xi,\epsilon,\eta}\omega = \mathcal{D}\xi + \sigma_+\eta + [\omega,\epsilon]_{\mathfrak{g}}\,, \qquad \delta_\epsilon \mathcal{A} = \mathcal{D}\epsilon\,, \tag{73}$$

where

$$\Omega_M^0 \otimes \mathbb{C}[y]^{\mathbb{Z}_2} \otimes \mathfrak{g} \ni \epsilon = \sum_{s=1}^\infty \frac{1}{(2s-2)!}\, y^{A(2s-2)}\, \epsilon_{A(2s-2)}\,, \tag{74}$$

is also promoted to a generating function for the gauge parameters associated with the higher spin $\mathfrak{g}$-valued fields making up $\mathcal{A}$. However, naïvely adapting the gauge transformations of $\Psi$ and $B$ by replacing them, as well as all gauge parameters, with the corresponding generating functions does not work: for instance, $\delta_\epsilon \Psi = [\Psi,\epsilon]_{\mathfrak{g}}$ does not belong to the same space as $\Psi$, which is $\Omega_M^0 \otimes \mathbb{C}[\bar{y}]^{\mathbb{Z}_2} \otimes \mathfrak{g}$, if $\epsilon \in \Omega_M^0 \otimes \mathbb{C}[y]^{\mathbb{Z}_2} \otimes \mathfrak{g}$ is the generating function (74) *and* the Lie bracket is understood as the $\mathbb{C}[y,\bar{y}]$-linear extension of that of $\mathfrak{g}$.

In order to gain insight into how to define these gauge transformations, we can look at the variation of the above action, which reads

$$\delta_{\xi,\eta,\epsilon}S[\Psi,\omega;\mathcal{A},\mathcal{B}] = \int_M p \circ \left\langle \delta_\epsilon \Psi, H \wedge \mathcal{D}\omega \right\rangle + p \circ \left\langle \delta_{\xi,\epsilon}\mathcal{B}, \mathcal{F} \right\rangle$$
$$- p \circ \left\langle \Psi, \left([\mathcal{F},\xi]_{\mathfrak{g}} + [\mathcal{D}\omega,\epsilon]_{\mathfrak{g}}\right) \wedge H \right\rangle. \tag{75}$$

As before, it seems natural to try and use the ad-invariance of the form on $\mathfrak{g}$ in order to compare the second line with the first one, and thereby fix the gauge transformation of $\Psi$ and $\mathcal{B}$. But to do so, we need to properly extend this invariance property to forms valued in $\mathbb{C}[y] \otimes \mathfrak{g}$ or $\mathbb{C}[\bar{y}] \otimes \mathfrak{g}$, that is to formulate it in terms of the $\mathbb{C}[y]$-linear extension of the Lie bracket of $\mathfrak{g}$ and the bilinear form $p \circ \left\langle -,- \right\rangle$. For this purpose, let us introduce the operation

$$\bullet : \mathbb{C}[\bar{y}] \otimes \mathbb{C}[y] \longrightarrow \mathbb{C}[\bar{y}]\,, \tag{76}$$

defined by

$$p\left(\psi \bullet f, g\right) = p\left(\psi, f \cdot g\right)\,, \tag{77}$$

or more concretely,

$$(\psi \bullet f)(\bar{y}) = \sum_{m,n \in \mathbb{N}} \frac{1}{m!\,n!}\, \bar{y}_{A(m)}\, \psi^{A(m)B(n)} f_{B(n)}\,, \tag{78}$$

for any $\psi \in \mathbb{C}[\bar{y}]$ and $f,g \in \mathbb{C}[y]$. In plain words, $\bullet$ is a representation of the commutative algebra $\mathbb{C}[y]$ on $\mathbb{C}[\bar{y}]$ seen as the dual[10] of $\mathbb{C}[y]$ via the pairing $p$. We can now write the identity

$$p \circ \left\langle \psi, [f,g]_{\mathfrak{g}} \right\rangle = p \circ \left\langle [\psi \overset{\bullet}{,} f]_{\mathfrak{g}}, g \right\rangle\,, \qquad f,g \in \mathbb{C}[y] \otimes \mathfrak{g}\,, \quad \psi \in \mathbb{C}[\bar{y}] \otimes \mathfrak{g}\,, \tag{79}$$

where we have introduced the notation $[-\overset{\bullet}{,}-]_{\mathfrak{g}}$ which should be understood as

$$[(\psi \otimes X)\overset{\bullet}{,}(f \otimes Y)]_{\mathfrak{g}} := (\psi \bullet f) \otimes [X,Y]_{\mathfrak{g}}\,, \qquad \psi \in \mathbb{C}[\bar{y}]\,, \quad f \in \mathbb{C}[y]\,, \quad X,Y \in \mathfrak{g}\,, \tag{80}$$

---

[10]Note that the space $\mathbb{C}[\bar{y}]$ of polynomials in $\bar{y}$ can be identified with the "restricted" dual of $\mathbb{C}[y]$. By "restricted" dual, we simply aim at pointing out the following subtlety: since the algebra $\mathbb{C}[y]$ is infinite-dimensional, one should be careful when talking about its dual space. Having in mind that this polynomial algebra is isomorphic to the symmetric algebra of the vector irrep of $\mathfrak{su}^*(4)$, i.e. $\mathbb{C}[y] \cong S(\square)$, and that the algebra of polynomials in $\bar{y}$ is isomorphic to the symmetric algebra of its conjugate representation, the following inclusion holds

$$\mathbb{C}[\bar{y}] \cong S(\blacksquare) \cong S(\square^*) \subset S(\square)^* \cong \mathbb{C}[y]^*\,.$$

In other words, the dual of the algebra of polynomials in $y$ *contains* the algebra of polynomials in $\bar{y}$ as a subalgebra, which is merely an instance of the standard fact that "the dual of an algebra of polynomials is an algebra of formal power series".

i.e. the extension of the Lie bracket by $\mathbb{C}[y, \bar{y}]$-linearity, composed with the bullet operation. We are now in position of writing the gauge transformations of $\Psi$ and $\mathcal{B}$,

$$\delta_{\xi, \eta, \epsilon} \mathcal{B} = -[\Psi \,\substack{\bullet \\ \bullet}\, \xi \wedge H]_{\mathfrak{g}} + [\mathcal{B} \,\substack{\bullet \\ \bullet}\, \epsilon]_{\mathfrak{g}}, \qquad \delta_\epsilon \Psi = [\Psi \,\substack{\bullet \\ \bullet}\, \epsilon]_{\mathfrak{g}}, \tag{81}$$

under which the action (70) is invariant, thanks to the invariance property (79) of $p \circ \langle -, - \rangle$. Note also that the gauge symmetry of $\omega$ is still reducible (on-shell), with

$$\mathring{\xi} = \mathcal{D}\zeta, \qquad \text{and} \qquad \mathring{\eta} = \sigma_- \zeta. \tag{82}$$

In summary, one can couple the tower of higher spin singletons described by the pair $(\omega, \Psi)$ to a "higher spin version" of a $\mathfrak{g}$-valued connection 1-form upon introducing a "higher spin version" of a BF theory.

**Adding a cubic term.** Given a 3-cocycle $\gamma \in \wedge^3 \mathfrak{g}^* \otimes \mathfrak{g}$ of the Lie algebra $\mathfrak{g}$ valued in its adjoint representation, one can also add another term to the action (70), namely

$$\int_M p \circ \langle \Psi, H \wedge \gamma(\mathcal{A}, \mathcal{A}, \mathcal{A}) \rangle, \tag{83}$$

where $\gamma$ is linearly extended to $\mathbb{C}[y] \otimes \Omega_M$. That $\gamma$ is a 3-cocycle ensures that the combination

$$\mathcal{G} := \mathcal{D}\omega - \tfrac{1}{6} \gamma(\mathcal{A}, \mathcal{A}, \mathcal{A}), \tag{84}$$

transforms as

$$\delta_\epsilon \mathcal{G} = [\mathcal{G}, \epsilon]_{\mathfrak{g}} + \gamma(\mathcal{F}, \mathcal{A}, \epsilon), \tag{85}$$

under the modified gauge transformations

$$\delta_\epsilon \omega = [\omega, \epsilon]_{\mathfrak{g}} + \tfrac{1}{2} \gamma(\mathcal{A}, \mathcal{A}, \epsilon). \tag{86}$$

Assuming further that the bilinear form $\langle -, - \rangle$ is *non-degenerate*, one can modify the gauge transformation of the field $\mathcal{B}$ as

$$\delta_\epsilon \mathcal{B} = [\mathcal{B} \,\substack{\bullet \\ \bullet}\, \epsilon]_{\mathfrak{g}} + \widetilde{\gamma}(\Psi \bullet H, \mathcal{A}, \epsilon), \tag{87}$$

where

$$\begin{aligned} \widetilde{\gamma} : \ &\mathfrak{g} \otimes (\mathfrak{g} \wedge \mathfrak{g}) \longrightarrow \mathfrak{g}, \\ &x \otimes (y \wedge z) \longmapsto \langle x, \gamma((-)^{\#}, y, z) \rangle, \end{aligned} \tag{88}$$

and $\# : \mathfrak{g}^* \longrightarrow \mathfrak{g}$ is defined by $\langle \alpha^{\#}, x \rangle \equiv \alpha(x)$ for any $\alpha \in \mathfrak{g}^*$ and $x \in \mathfrak{g}$. Explicitly, if we choose a basis $\{e_a\}$ of $\mathfrak{g}$, the components of $\widetilde{\gamma}$ read

$$\widetilde{\gamma}_{abc}{}^d = \kappa_{ai} \gamma_{jbc}{}^i \kappa^{jd}, \tag{89}$$

where $\kappa_{ab} := \langle e_a, e_b \rangle$ are the components of the invariant bilinear form, and $\kappa^{ab}$ those of its inverse (i.e. $\kappa_{ac} \kappa^{bc} = \delta_a^b$). This definition implies the identity

$$\langle w, \gamma(x, y, z) \rangle = \langle \widetilde{\gamma}(w, x, y), z \rangle, \qquad \forall w, x, y, z \in \mathfrak{g}, \tag{90}$$

which in turn ensures that the action

$$S[\Psi, \omega; \mathcal{A}, \mathcal{B}] = \int_M p \circ \langle \Psi, H \wedge \mathcal{G} \rangle + p \circ \langle \mathcal{B}, \mathcal{F} \rangle, \tag{91}$$

is invariant under the modified transformations (85) generated by $\epsilon$, as well as the gauge transformations generated by $\xi$ and $\eta$ which are unaffected by the addition of the cubic term (83).[11]

---

[11]Let us stress that the possibility of adding such a cubic term is a specificity of the type of theory under consideration here, that is, involving chiral fields in 6 dimensions.

**Non-linearities in the zero-forms.** Finally, let us conclude this section by pointing out that the free action (31) is of *presymplectic AKSZ-type* [131, 132], meaning they are obtained by the AKSZ construction [133] with a presymplectic $\mathcal{Q}$-manifold as its target space instead of a symplectic one as is usually assumed. This is reflected in the fact

$$p\big(\Psi, H \wedge -\big) : \Omega_M^3 \otimes \mathbb{C}[y] \longrightarrow \Omega_M^6 \,, \tag{92}$$

has a kernel (as a consequence of $H_{AA} \wedge e_{A,B} = 0$), which ensures that the free action (70) is gauge invariant. As a consequence, the sum of the free actions for spin $s \geq 1$ singletons can be deformed to

$$S[\Psi, \omega] = \int_M p\big(\Theta(\Psi), H \wedge \nabla \omega\big), \qquad \text{where} \qquad \Omega_M^0 \otimes \mathbb{C}[\bar{y}] \ni \Theta(\Psi) = \Psi + \mathcal{O}(\Psi^2), \tag{93}$$

is a $\mathbb{C}[\bar{y}]$-valued 0-form which admits an expansion in powers of $\Psi$ and starts with a linear term (so as to recover the free theory). Note that the presymplectic potential $\Theta(\Psi)$ for four-dimensional higher spin gravity was constructed up to second order in the zero-form in [134, Sec. 6], building on the result of [135] for the free theory.

On top of any polynomial in $\Psi$, one can construct examples using the operator

$$\mathcal{I}_4 = \varepsilon_{ABCD}\,\varepsilon_{ABCD}\,\frac{\partial^2}{\partial \bar{y}_A \partial \bar{y}_A} \otimes \frac{\partial^2}{\partial \bar{y}_B \partial \bar{y}_B} \otimes \frac{\partial^2}{\partial \bar{y}_C \partial \bar{y}_C} \otimes \frac{\partial^2}{\partial \bar{y}_D \partial \bar{y}_D}\,, \tag{94}$$

which acts on four $\Psi$ as

$$m_4 \circ \mathcal{I}_4(\Psi^{\otimes 4}) = \varepsilon_{ABCD}\,\varepsilon_{ABCD} \sum_{s_k \in 2\mathbb{N}} \frac{1}{s_1! s_2! s_3! s_4!}\,\Psi^{AAE(s_1)}\Psi^{BBE(s_2)}\Psi^{CCE(s_3)}\Psi^{DDE(s_4)}\,\bar{y}_{E(s_1+s_2+s_3+s_4)}\,, \tag{95}$$

where we have also introduced the operator $m_4 : \mathbb{C}[\bar{y}]^{\otimes 4} \to \mathbb{C}[\bar{y}]$, denoting the multiplication of (four) polynomials of $\bar{y}_A$.[12] In other words, any $\Theta$ of the form

$$\Theta(\Psi) = \Psi + \sum_{k \geq 2} c_k \Psi^k + \sum_{l \geq 1} \tilde{c}_l\, m_4 \circ \mathcal{I}_4^k(\Psi^{\otimes 4})\,, \tag{96}$$

for $c_k, \tilde{c}_k$ some (arbitrary real) coefficients, can be used to define the interacting action (93).

As a side comment, let us note that nonlinearities in the zero-forms can also be introduced in the presence of a BF-system, provided one is given a $(\mathbb{C}[y] \otimes \mathfrak{g})$-equivariant map

$$\Theta : \bigoplus_{n \in \mathbb{N}} \big(\mathbb{C}[\bar{y}] \otimes \mathfrak{g}\big)^{\otimes n} \longrightarrow \mathbb{C}[\bar{y}] \otimes \mathfrak{g}\,, \tag{97}$$

so that for $\Psi \in \Omega_M^0 \otimes \mathbb{C}[\bar{y}] \otimes \mathfrak{g}$ and transforming as in (81), the polynomial $\Theta(\Psi)$ transforms similarly,

$$\delta_\epsilon \Psi = [\Psi \,\overset{\bullet}{,}\, \epsilon]_{\mathfrak{g}} \qquad \Longrightarrow \qquad \delta_\epsilon \Theta(\Psi) = [\Theta(\Psi) \,\overset{\bullet}{,}\, \epsilon]_{\mathfrak{g}}\,. \tag{98}$$

In this case, we can replace $\Psi$ for $\Theta(\Psi)$ in the action (70) discussed above, i.e.

$$S[\Psi, \omega; \mathcal{A}, \mathcal{B}] = \int_M p \circ \big\langle \Psi, H \wedge \mathcal{D}\omega \big\rangle + p \circ \big\langle \mathcal{B}, \mathcal{F} \big\rangle\,, \tag{99}$$

which becomes gauge-invariant under

$$\delta_{\xi,\epsilon,\eta}\omega = \mathcal{D}\xi + \sigma_+ \eta + [\omega, \epsilon]_{\mathfrak{g}}\,, \qquad \delta_\epsilon \mathcal{A} = \mathcal{D}\epsilon\,, \qquad \delta_\epsilon \Psi = [\Psi, \epsilon]_{\mathfrak{g}}\,, \tag{100}$$

---

[12]Note that if one only keeps the spin 1 sector, that is $\Psi = \frac{1}{2}\Psi^{AA}\bar{y}_A\bar{y}_A$, $m_4 \circ \mathcal{I}_4$ is the invariant considered in [123, Sec. 4.3] for the democratic formulation of non-linear theories for chiral 2-forms.

provided the field $\mathcal{B}$ transforms as

$$\delta_{\xi,\eta,\epsilon}\mathcal{B} = -[\Theta(\Psi)\overset{\bullet}{,}\xi\wedge H]_{\mathfrak{g}} + [\mathcal{B}\overset{\bullet}{,}\epsilon]_{\mathfrak{g}}\,,\tag{101}$$

by the same mechanism as the one discussed in the previous section.

Lastly, let us point out that one can also build simple current-type interactions. To do so, suppose that we have a 1-*form* $J^{A(2s)}$ which verifies

$$\nabla J^{A(2s-2)BB}\wedge H_{BB}\approx 0\,,\tag{102}$$

upon using the equations of motions (hence our use of the symbol $\approx$ in the above equation). One can then add the functional

$$\int_M J^{A(2s)}\wedge H_{AA}\wedge\omega_{A(2s-2)}\,,\tag{103}$$

to the free actions of all singletons involved, i.e. that of spin-$s$, as well as all those involved in the definition of $J^{A(2s)}$, and thereby obtain a current-type interaction, which is *on-shell* gauge-invariant. Indeed, a simple computation shows that such a term is invariant under the shift symmetry of $\omega$ as a consequence of the property (16), and invariant under the differential gauge symmetry of $\omega$ *on-shell*, thanks to the condition (102). Such interaction terms could be used as a starting point to define other interacting theories, possibly involving additional fields not discussed here, together with a number of singletons (potentially a finite one, in the spirit of the current interactions discussed in [130] or the more general analysis of finite-spectrum higher spin theories in four dimensions given in the recent [136]).

Having a 1-form conserved current as described in the previous paragraph would constitute an example of a 1-form symmetry, the simplest instance of higher-form symmetries [137, 138] (see e.g. [139, 140] for an introduction). Such symmetries are usually presented in terms of a 2-form current $\mathfrak{J}_{[2]}\in\Omega_M^2$, which is conserved on-shell in the sense that $\mathrm{d}*\mathfrak{J}_{[2]}\approx 0$. This type of current naturally couples to a 2-form gauge field $\mathfrak{O}_{[2]}\in\Omega_M^2$ via $\int_M\mathfrak{O}_{[2]}\wedge *\mathfrak{J}_{[2]}$, in *any dimensions*. In our 6*d* example here, the 2-form current is the Hodge dual of the 4-form $J^{A(2s-2)BB}\wedge H_{BB}\sim *\mathfrak{J}_{[2]}$, so that the functional (103) is merely a re-writing of the expected current coupling term.

## 3 Arbitrary even dimensions

The $d=4$ case [20] and the $d=6$ case discussed in the previous section are so similar that it seems reasonable to expect that the same approach can be successfully followed in arbitrary *even dimensions*, $d=2r$. Taking a step back, one can notice that the field content of the action (31) in four and six dimensions is captured by the following objects:

- A 0-form

$$\Psi_{\pm}^{a_1(s),a_2(s),\ldots,a_r(s)}\qquad\longleftrightarrow\qquad \boxed{\phantom{xxx}}\,,\tag{104}$$

    taking values in the *self-dual or anti-self-dual* Lorentz irrep[13] (respectively denoted by a $+$ or $-$ subscript) corresponding to a rectangular Young diagram of length $s$ and maximal height, that is $\frac{d}{2}=r$;

---

[13]Self-duality for such a 0-forms means that it verifies

$$\tfrac{1}{r!}\,\epsilon^{a_1\ldots a_r}{}_{b_1\ldots b_r}\,\Psi_{\pm}^{a_1(s-1)b_1,a_2(s-1)b_2,\ldots,a_r(s-1)b_r} = \pm i^{r-2\left[\frac{r}{2}\right]}\,\tfrac{1}{r!}\,\Psi^{a_1(s),a_2(s),\ldots,a_r(s)}\,,$$

with $\left[\frac{r}{2}\right]$ the integer part of $\frac{r}{2}$, i.e. its "fiberwise Hodge dualisation" in *any* column is proportional to $\pm 1$ (resp. $\pm i$) times itself for $r$ even (resp. odd).

- A $\frac{d-2}{2} = (r-1)$-form taking values in

$$\omega^{(-1)^r\pm}_{a_1(s-1),\dots,a_r(s-1)} \qquad \longleftrightarrow \qquad \boxed{\begin{array}{c} s-1 \\ \\ \\ \end{array}}_{(-1)^r\pm}, \qquad (105)$$

taking values in the *(anti-)self-dual* Lorentz irrep characterised by another rectangular Young diagram, of (maximal) height $r$ and length $s-1$, i.e. with one column less than the 0-form.

For the sake of readability, we will hereafter suppress the $\pm$ sub/superscript of $\Psi$ and $\omega$, and will assume that $\Psi$ is self-dual while $\omega$ is takes values in a self-dual irrep if $r$ is even, and anti-self-dual if $r$ is odd. We can then consider the action

$$S[\Psi,\omega] = \int_M \Psi^{a_1(s),\dots,a_r(s)} e_{a_1} \wedge \cdots \wedge e_{a_r} \wedge \nabla \omega_{a_1(s-1),\dots,a_r(s-1)}, \qquad (106)$$

where $\nabla$ is a connection of constant curvature,

$$\nabla^2 = e^a \wedge e^b \, \rho(M_{a,b}), \qquad (107)$$

where $M_{a,b}$ are the generators of the Lorentz algebra and $\rho$ the representation in which the fields acted upon fall.[14] Note that the $r$-form $e_{a_1} \wedge \cdots \wedge e_{a_r}$ being contracted with a self-dual tensor $\Psi$, only its self-dual part—if $r$ is even—or anti-self-dual part—if $r$ is odd—appears in the integrand.[15] The previous action is invariant under the gauge transformations

$$\delta_{\xi,\eta}\omega_{a_1(s-1),\dots,a_r(s-1)} = \nabla\xi_{a_1(s-1),\dots,a_r(s-1)} + e_{a_r} \wedge \eta_{a_1(s-1),\dots,a_{r-1}(s-1),a_r(s-2)} + (\dots), \qquad (108)$$

where the $(\dots)$ denote additional terms needed in order to project the wedge product of a vielbein with the $(r-2)$-form

$$\eta_{a_1(s-1),\dots,a_{r-1}(s-1),a_r(s-2)} \qquad \longleftrightarrow \qquad \boxed{\begin{array}{c} s-1 \\ \\ \end{array}}_{s-2}, \qquad (109)$$

onto the symmetry of the rectangular diagram (105), and make it traceless.

The *algebraic* gauge symmetry generated by $\eta$ allows one to gauge away certain "unwanted" components of the $\frac{d-2}{2}$-form $\omega$, which are represented by the ellipsis in bracket below,

$$r-1\left\{\boxed{\phantom{x}} \otimes \boxed{\begin{array}{c} s-1 \\ \\ \end{array}} \cong \boxed{\begin{array}{c} s \\ \\ s-1 \end{array}} \oplus \big[\cdots\big]. \qquad (110)$$

From the point of view of the above tensor product, the irreps comprised in $\big[\cdots\big]$ are *traces*, and can be written as

$$\big[\cdots\big] \cong \bigoplus_{k=2}^{r} (r-k)\left\{\boxed{\phantom{x}} \underset{\text{LR}}{\otimes} \overbrace{\boxed{\begin{array}{c} s-1 \\ \\ \end{array}}}^{s-1}, \qquad (111)$$

where $k-1$ boxes have been remove along the rightmost column of the diagrams on the right of the symbol $\underset{\text{LR}}{\otimes}$, which denotes the Littlewood–Richardson rule for the tensor product of two

---

[14]Here again we implicitly normalise the curvature.

[15]Recall that in dimension $d = 2r$, the wedge product of two self-dual $r$-forms vanish identically for $r$ odd and are proportional to the volume form for $r$ even, while the wedge product of a self-dual and an anti-self-dual $r$-form identically vanishes for $r$ even and is proportional to the volume form for $r$ odd.

Young diagrams (see e.g. [42, Sec. 4] for a pedagogical introduction). These diagrams are recovered among the irreducible components of the $\frac{d-4}{2}$-form $\eta$, which can again be sorted into two pieces,

$$
r-2\left\{\boxed{\phantom{x}}\otimes\begin{array}{c} \overset{s-1}{\boxed{\phantom{xxxx}}} \\ \underset{s-2}{\boxed{\phantom{xxxx}}}\end{array}\right. \cong [\cdots]\oplus\{\cdots\}, \tag{112}
$$

where the extra term $\{\cdots\}$ can be expressed in a similar manner as (111). Though we will not give the precise decomposition of this term here, one can check that it is contained in the tensor product

$$
r-3\left\{\boxed{\phantom{x}}\otimes\begin{array}{c} \boxed{\phantom{xxxx}} \\ \underset{s-2}{\boxed{\phantom{xx}}}\end{array}\right., \tag{113}
$$

corresponding to the $\mathfrak{so}(1,d-1)$-irrep content of the reducibility parameter of the gauge symmetry (108) of $\omega$. As it turns out, this reducibility parameter contains again additional terms on top of those included in $\{\cdots\}$, which are themselves contained in a $(r-4)$-form valued in the diagram whose $r-3$ first row have length $s-1$ and the last 3 have length $s-2$ (i.e. the diagram obtained by removing three boxes along the rightmost column of the rectangular diagram of height $r$ and length $s-1$). Indeed, the latter is nothing but the second stage reducibility parameter of the gauge transformations considered here. This process goes on: the gauge symmetry admits several stages of reducibility, whose parameters are obtained by reducing simultaneously the form degree and the number of boxes in the rightmost column of the Lorentz diagram by one at each stage. The Lorentz irreps appearing in the decomposition of a parameter at a given stage of reducibility always split into two disjoint sets, each one contained in the decomposition of the parameters at either the previous or the next stage. This ensures that $\omega$ only propagates the diagram obtained by removing one box in the lower right corner of the rectangular diagram of height $r$ and length $s$.

The equations of motion resulting from the variation of the above action read

$$
\mathbb{P}_s\left(H^{(-1)^r}_{a_1,\ldots,a_r}\wedge\nabla\omega_{a_1(s-1),\ldots,a_r(s-1)}\right)\approx 0, \qquad\text{and}\qquad \nabla\Psi^{a_1(s-1)b_1,\ldots,a_r(s-1)b_r}H^{(-1)^r}_{b_1,\ldots,b_r}\approx 0, \tag{114}
$$

where $\mathbb{P}_s$ denotes the projection onto the diagram which is a $r\times s$ rectangle, and $H^{\pm}_{a_1,\ldots,a_r}$ denotes the self-dual or anti-self-dual part of the wedge product $e_{a_1}\wedge\cdots\wedge e_{a_r}$ of $r$ vielbeins. Writing the self-dual and anti-self-dual part of $\nabla\omega$ as,

$$
\nabla\omega_{a_1(s-1),\ldots,a_r(s-1)}=H^{b_1,\ldots,b_r}_{+}C^{(-1)^r}_{a_1(s-1)b_1,\ldots,a_r(s-1)b_r}+H^{b_1,\ldots,b_r}_{-}C^{-(-1)^r}_{a_1(s-1)b_1,\ldots,a_r(s-1)b_r}, \tag{115}
$$

where the sign of the superscript on the 0-forms $C$ denotes its self-dual or anti-self-dual character, and using the fact that

$$
H^{\mp}_{a_1,\ldots,a_r}\wedge H^{(-1)^r\pm}_{b_1,\ldots,b_r}=0, \tag{116}
$$

the first equation implies that $\nabla\omega$ only contains an anti-self-dual piece, i.e. on-shell

$$
\nabla\omega_{a_1(s-1),\ldots,a_r(s-1)}\approx H^{b_1,\ldots,b_r}_{-}C^{-(-1)^r}_{a_1(s-1)b_1,\ldots,a_r(s-1)b_r}, \tag{117}
$$

and is therefore determined by a 0-form which is anti-self-dual for $r$ even, and self-dual for $r$ odd. In other words, it describes a singleton of spin $s$ and of negative/positive chirality for $r$ even/odd. The second equation of motion, that of $\Psi$, allows one to parameterise its first derivative, by terms lying in the kernel of the map which consists in contracting with the $r$-form $H_{a_1,\ldots,a_r}$, and repeat this process for the 0-forms introduced by imposing Bianchi identities. This introduces new 0-forms, and repeating these steps *ad infinitum*, one obtains a system of first order differential equations for infinitely many 0-forms,[16] describing the free

---

[16]This procedure is known as "unfolding" in the context of higher spin theories.

propagation of a singleton of spin $s$ described by the field $\Psi$, and hence of positive chirality. As a consequence, we find that the action (106) discussed in this section describes a pair of singleton of spin $s$ and of *opposite/same chirality* in $d = 2r$ dimensions for $r$ even/odd. This is consistent with the $4d$ case treated in [20] and the $6d$ case discussed in the previous section.

Note that free actions for conformal fields of arbitrary mixed-symmetries have been studied in details [141], and for higher spin singletons in particular [142, 143], however in a slightly different form than the ones we consider here.

## 3.1 Adding colour

Now that we have a free action in arbitrary even dimensions of the same form as (31) in $6d$, it becomes straightforward to extend the (minimal) coupling to a $\mathfrak{g}$-valued connection 1-form described in the previous section: nothing in this construction relies crucially on the fact that $M$ is six-dimensional. The simplicity of the free field description is due to the fact that higher spin singletons carry a particular irrep of $\mathfrak{so}(1, d-1)$, namely that corresponding to a *rectangular* Young diagram, of *maximal* height. Moreover, the BF fields in (55) *do not* mix singletons of different spins, but couple to them individually. In other words, we can write the action

$$S[\Psi, \omega; A, B] = \int_M \left\langle \Psi^{a_1(s),\ldots,a_r(s)}, H_{a_1,\ldots,a_r} \wedge D\omega_{a_1(s-1),\ldots,a_r(s-1)} \right\rangle + \left\langle B, F \right\rangle, \tag{118}$$

where

- We used the notation $H_{a_1,\ldots,a_r} := e_{a_1} \wedge \cdots \wedge e_{a_r}$ with indices suppressed;

- The 0-form $\Psi^{a_1(s),\ldots,a_r(s)}$, the $\frac{d-2}{2}$-form $\omega_{a_1(s-1),\ldots,a_r(s-1)}$, the 1-form $A$ and the $(d-2)$-form $B$ all take values in a Lie algebra $\mathfrak{g}$;

- We again denoted the covariant derivative with respect to the $\mathfrak{g}$-valued gauge field $A$ as $D := \nabla + [A, -]_\mathfrak{g}$, and its curvature as $F := dA + \frac{1}{2}[A, A]_\mathfrak{g}$.

The action (118) is readily verified to be invariant under

$$\begin{aligned} \delta_{\xi,\epsilon}\omega &= D\xi + [\omega, \epsilon]_\mathfrak{g}, & \delta_\epsilon\Psi &= [\Psi, \epsilon]_\mathfrak{g}, \\ \delta_\epsilon A &= D\epsilon, & \delta_{\xi,\epsilon}B &= \mathbb{P}_0\big([\Psi, H \wedge \xi]_\mathfrak{g}\big), \end{aligned} \tag{119}$$

where $\xi$ is a $(r-1)$-form whose indices, as well as those of $\omega$ and $\Psi$, have been suppressed for simplicity, $\epsilon$ is a $\mathfrak{g}$-valued 0-form, in the trivial representation of the "fiber" Lorentz algebra (i.e. not carrying any Latin indices) and $\mathbb{P}_0$ denotes the projection onto the trivial Lorentz irrep. In other words, the indices of the 0-form $\Psi$ are completely contracted with those of $H$ and $\xi$. On top of that, the action is also invariant under the transformations

$$\delta_\eta\omega = \sigma_+\eta, \tag{120}$$

where $\sigma_\pm$ are defined as the piece of the action of the transvection generators $P_a$ which relates $\mathfrak{so}(1, d-1)$ irreps making up the $\mathfrak{so}(2, d-1)$ irrep labelled by a $(r+1) \times (s-1)$ diagram by respectively adding or substracting a box in their Young diagram. Schematically, these

operators act as follows

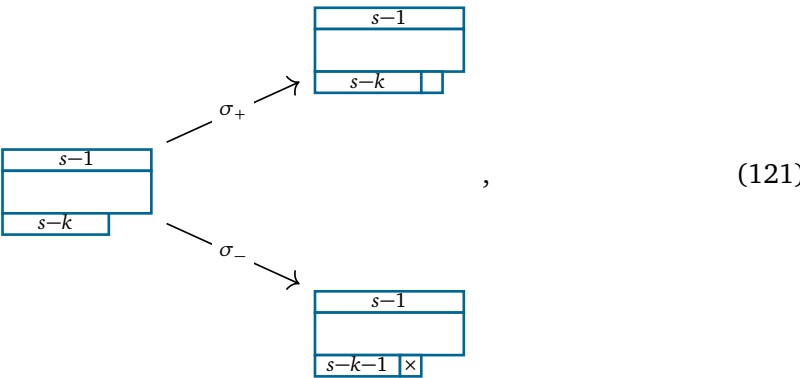

$$\tag{121}$$

where all diagrams here correspond to irreps of Lorentz algebra $\mathfrak{so}(1, d-1)$ and hence are of height $r$. As before, the gauge transformations generated by $\eta$ are meant to ensure that $\omega$ propagetes the correct number of degrees of freedom on-shell.

In principle, one could push further and construct a counterpart of the higher spin extension of the above theory in arbitrary even dimensions: to reproduce the $6d$ case, all one needs is a commutative algebra to tensor with the Lie algebra $\mathfrak{g}$, thereby replacing the algebra of (even) polynomials in $y^A$ used above, $\mathbb{C}[y]^{\mathbb{Z}_2}$. As explained previously, the latter decomposes into a direct sum of finite-dimensional $\mathfrak{su}^*(4)$-modules, namely

$$\mathbb{C}[y]^{\mathbb{Z}_2} \cong \bigoplus_{s=1}^{\infty} \boxed{\phantom{xx}2s-2\phantom{xx}}, \tag{122}$$

where each one-row diagram of $\mathfrak{su}^*(4)$ corresponds to a three-row rectangular diagram of $\mathfrak{so}(1, 5)$, whose length is half of that of the $\mathfrak{su}^*(4)$ one. This suggests looking for an algebra in $d = 2r$ dimensions whose decomposition consists of all rectangular $r \times (s-1)$ Young diagrams of $\mathfrak{so}(1, d-1)$ with $s \geq 1$,

$$\mathfrak{hs}_{\pm}^{\mathrm{ab}} \cong \bigoplus_{s=1}^{\infty} \boxed{\phantom{xxxx}}_{\pm}^{s-1}, \tag{123}$$

and consider it as a commutative associative algebra. One could think doing so by defining the multiplication as the *Cartan product*[17] of irreps of $\mathfrak{so}(1, d-1)$, which indeed defines an associative and commutative algebra [144]. Explicitly realising this algebra would however involve the use of projection operators for these rectangular diagrams, which quickly become technically difficult to write down (see e.g. [145] for recent work on building traceless projectors for arbitrary Young diagrams).

## 3.2 Some generalities

Recall the first order formulation of Yang–Mills theory,

$$S[\mathrm{Y}, A] = \int_M \left\langle \mathrm{Y}, F - \tfrac{1}{2}\, g_{\mathrm{YM}} * \mathrm{Y} \right\rangle_{\mathfrak{g}}, \tag{124}$$

---

[17]The Cartan product of two finite-dimensional highest weight representations of a semisimple Lie algebra is simply the projection onto the irrep, in their tensor product, whose highest weight is the sum of their two highest weights. It is also the irrep with the maximal dimension in the tensor product. In terms of the two Young diagrams associated with the highest weights, it simply corresponds to gluing/concatenating them side-by-side (i.e. the length of $k$th row of the new diagram is the sum of the length of the $k$th row of the two original ones).

where $F$ is the field strength of the Yang–Mills field $A \in \Omega_M^1 \otimes \mathfrak{g}$ for $\mathfrak{g}$ a Lie algebra with $\langle -, - \rangle_{\mathfrak{g}}$ its invariant bilinear form, $g_{\text{YM}}$ the Yang–Mills coupling constant, $*$ denotes the Hodge dual on the $d$-dimensional manifold $M$, and $\Upsilon \in \Omega_M^{d-2} \otimes \mathfrak{g}$ is a Lagrange multiplier. Integrating out $\Upsilon$, one recovers the usual, second order, Yang–Mills action. In $4d$, one has the possibility of "truncating" this action by keeping only the (say) self-dual part of $\Upsilon$ and neglecting the term quadratic in it (by taking the weak coupling limit $g_{\text{YM}} \to 0$). Upon writing $\Upsilon = \Psi^{\alpha\alpha} H_{\alpha\alpha}$ with $\Psi^{\alpha\alpha} \in \Omega_M^0 \otimes \mathfrak{g}$ the components of the self-dual part of the field strength, and $H_{\alpha\alpha}$ a basis of self-dual 2-forms, written in terms of $\mathfrak{sl}(2,\mathbb{C})$-irreps, or two-component spinors, one ends up with the action

$$S[\Psi, A] = \int_M \left\langle \Psi^{\alpha\alpha}, H_{\alpha\alpha} \wedge F \right\rangle_{\mathfrak{g}}, \tag{125}$$

considered as the starting point to obtain an higher spin extension of self-dual Yang–Mills [20].

Since we are interested in theories containing at least a 2-form gauge fields, we can turn our attention to a class of higher gauge theories (see e.g. [146, 147]) based on two-term $L_\infty$-algebra, or Lie 2-algebra. For simplicity, let us focus on a *minimal*[18] Lie 2-algebra, which boils down to the data of a(n ordinary) Lie algebra $\mathfrak{g}$ in degree 0, a module $V$ over it in degree $-1$, and a Chevalley–Eilenberg 3-cocycle $\gamma \in \wedge^3 \mathfrak{g}^* \otimes V$ for $\mathfrak{g}$ valued in $V$. The Lie bracket on $\mathfrak{g}$ and the representation of $\mathfrak{g}$ on $V$ assemble into a binary bracket (of degree 0) on the graded vector space $\mathfrak{g} \oplus V$ (hence concentrated in degrees 0 and $-1$), while the cocycle defines a ternary bracket (of degree $-1$). The gauge fields associated with such an algebraic structure consists of a $\mathfrak{g}$-valued 1-form $A \in \Omega_M^1 \otimes \mathfrak{g}$ and a $V$-valued 2-form $\omega \in \Omega_M^2 \otimes V$. The gauge transformations of these fields are given by

$$\delta_{\epsilon,\xi} A = \mathrm{d}\epsilon + [A, \epsilon]_{\mathfrak{g}}, \qquad \delta_{\epsilon,\xi} \omega = \mathrm{d}\xi + \rho(A)\xi - \rho(\epsilon)\omega + \tfrac{1}{2}\gamma(A, A, \epsilon), \tag{126}$$

for $\epsilon \in \Omega_M^0 \otimes \mathfrak{g}$ and $\xi \in \Omega_M^1 \otimes V$. The curvature of the 1-form gauge field $A$ is unchanged (meaning, it takes the same form as in the Yang–Mills case), while the curvature of the 2-form is defined, and transforms, as

$$G = \mathrm{d}\omega + \rho(A)\omega - \tfrac{1}{6}\gamma(A, A, A), \qquad \delta_{\epsilon,\xi} G = -\rho(\epsilon)G + \rho(F)\xi + \gamma(F, A, \epsilon). \tag{127}$$

The important difference with respect to the usual Yang–Mills field strength is that the curvature of the 2-form gauge field $\omega$ *does not* transform into *itself* under (either types of) the gauge transformations (unless, say $F = 0$). As a consequence, the naïve guess for an action for $\omega$ that would be

$$S_{\text{YM–like}}[A, \omega] = \tfrac{1}{2} \int_M \left\langle G, *G \right\rangle_V, \tag{128}$$

where $\langle -, - \rangle_V$ is a $\mathfrak{g}$-invariant[19] symmetric bilinear form on $V$, is not gauge invariant. Instead, its gauge variation takes the form

$$\delta_{\epsilon,\xi} S = \int_M \left\langle \rho(F)\xi + \gamma(F, A, \epsilon), *G \right\rangle_V, \tag{129}$$

---

[18]A minimal $L_\infty$-algebra is one for which the differential is trivial. Recall that a Lie 2-algebra is a $L_\infty$-algebra structure on a two-term complex $V \xrightarrow{\partial} \mathfrak{g}$ concentrated in degrees $-1$ and $0$. This last restriction on the grading implies that the only non-trivial brackets (on top of the differential) are of arity two and three. When the differential is trivial, such a structure amounts to the data summarised above, namely a Lie algebra structure on $\mathfrak{g}$, a module structure on $V$, and a $V$-valued 3-cocycle. Whenever the differential is non-trivial however, the $L_\infty$-algebra structure does not consists in these usual Lie theoretic objects, as for starter the Jacobiator of the binary bracket on $\mathfrak{g}$ no longer vanishes but is equal to the composition $\partial \circ \gamma$, so that $\mathfrak{g}$ is not a Lie algebra.

[19]Meaning that it obeys $\langle \rho(x)v, w \rangle_V + \langle v, \rho(x)w \rangle_V = 0$ for any $x \in \mathfrak{g}$ and $v, w \in V$.

i.e. gauge invariance of the naïve guess (128) is obstructed by terms proportional to the field strength $F$. This suggests the addition of a BF term as a way of restoring gauge invariance (see e.g. [148, Sec. 8]): indeed, the variation of the functional

$$S_{\text{BF}}[B, A] = \int_M \left\langle B, F \right\rangle, \tag{130}$$

with $B \in \Omega_M^{d-2} \otimes \mathfrak{g}^*$ a $(d-2)$-form valued in the *dual* of $\mathfrak{g}$, compensate the variation (129) of the naïve guess (128), provided that the field $B$ transform as

$$\delta_{\epsilon, \xi} B = [B, \epsilon]_{\mathfrak{g}}^* - \left\langle \rho(-)\xi + \gamma(-, A, \epsilon), *G \right\rangle_V, \tag{131}$$

where $[-, -]_{\mathfrak{g}}^*$ denotes the coadjoint action of $\mathfrak{g}$ on its dual $\mathfrak{g}^*$ and the dash $(-)$ in $\rho$ and $\gamma$ denotes an empty entry for elements of $\mathfrak{g}$ (since the whole expression is valued in $\mathfrak{g}^*$).

At this point, we can easily find a first order formulation for the gauge invariant action obtained above, namely

$$S[A, \omega; Y, B] = \int_M \left\langle Y, G - \tfrac{1}{2} * Y \right\rangle_V + \left\langle B, F \right\rangle, \tag{132}$$

i.e. we have introduced an auxiliary field $Y \in \Omega_M^{d-3} \otimes V$ such that its integration will set it to be $*G$, the Hodge dual of the curvature of the 2-form $\omega$. Now fixing the dimension to $d = 6$, we can split the 3-forms into self-dual and anti-self-dual ones, and as in the $4d$ case previously discussed, keep *only the self-dual part* of $Y$ which we write as $\Psi^{AA} H_{AA}$ with $H_{AA}$ the basis of self-dual 3-forms in $\mathfrak{su}^*(4)$ notation introduced in Section 2.1. After doing so (and neglecting the term quadratic in $Y$), one ends up with the action

$$S[A, \Psi; \omega, B] = \int_M \left\langle \Psi^{AA}, H_{AA} \wedge G \right\rangle_V + \left\langle B, F \right\rangle, \tag{133}$$

which constitute the starting point of the higher spin extension proposed here.

## 4 Discussion

Six-dimensional spacetime is known to offer interesting possibilities for "exotic"—in terms of field content—theories, including mixed-symmetry fields. In this note, we discussed the case of higher spin singletons, which are examples of such mixed-symmetry fields having some very peculiar properties—for one thing they are conformal. Thanks to the simple formulation of these fields afforded by the isomorphism $\mathfrak{so}(1, 5) \cong \mathfrak{sl}(2, \mathbb{H}) \cong \mathfrak{su}^*(4)$, we were able to derive a couple of examples of interacting theories (which however break conformal invariance) without encountering real difficulties or subtleties. One can expect that other examples should be possible to construct. For instance, considering that the field content involve 2-forms, one may expect the existence of higher-form currents which could serve as the basis for building current interactions as mentioned in the end of Section 2.2, and thereby provide new examples of higher-form symmetries [137, 138] (see e.g. [139, 140] for recent reviews).

Let us conclude this note by listing, in no particular order, a few interesting directions and questions, that we would hope to explore next.

- Considering that the contraction of chiral higher spin gravity that is the higher spin extension of self-dual Yang–Mills seems to find some simple counterpart in $6d$, as presented in this note, it is worth asking whether or not the same is also true of the full theory. In other words, does a counterpart of chiral higher spin gravity exist in $6d$? If so, it would not only be interesting as another example of complete higher spin theory, but also because it would involve mixed-symmetry / higher forms gauge fields, whose interactions are notoriously difficult to construct, and scarce in examples.

- As stated in the introduction, chiral higher spin gravity in $4d$ admits a contraction to a higher spin extension of self-dual gravity. Added to the fact that the singleton of spin $s = 2$ may be used for an exotic formulation of gravity, one may expect that a $6d$ counterpart also exists. It was recently proposed that the $4d$ higher spin extension of self-dual gravity can be obtained from a theory in $6d$, invariant under diffeomorphisms for the total six-dimensional space [29]. One could speculate about the possibility of finding a counterpart of this result, namely deriving the putative $6d$ version of higher spin self-dual gravity just mentioned from *ten dimensions*, by extending spacetime with the four-dimensional spinor space (for which the additional variables $y^A$ introduced to define generating functions would be coordinates).

- Another intriguing possibility would be to obtain an higher spin version of the Penrose–Ward correspondence in $6d$, in a similar manner as the one recently derived in $4d$ [25, 26]. As discussed in [111, Sec. 5], the related problem of defining a "twistor transform" which establishes a bijection between solutions of the massless free equation for a spin $s$ singleton in $6d$ and some cohomology class on twistor space is not devoid of difficulties and subtleties (see also [112, 149] and references therein). A closely related issue would be to derive an action for theories considered here in twistor space, following the works [23, 24, 26].

## Acknowledgments

I am grateful to Evgeny Skvortsov for numerous discussions during the completion of this work, and to Victor Lekeu for enlightening discussions on chiral $p$-forms, and I am indebted to both of them for valuable feedback on a previous version of this paper. I am also thankful to Thanasis Chatzistavrakidis and Sylvain Lavau for discussions on chiral $p$-forms, tensor hierarchies and topics related to the content of this note. Finally, I am grateful to the anonymous Referees for their careful reading and insightful suggestions, which have contributed to improving my understanding of the subject as well as the quality of this manuscript.

**Funding information** This work was supported by the European Union's Horizon 2020 research and innovation programme under the Marie Skłodowska Curie grant agreement No 101034383, as well as from the European Research Council (ERC) under the European Union's Horizon 2020 research and innovation programme grant agreement No 10100255.

## A   Partially-massless extension

The higher spin extension of self-dual Yang–Mills in $4d$ [20] admits a "partially-massless" counterpart [130], that is a theory whose spectrum consists of fields of arbitrary spin and depth, valued in a Lie algebra $\mathfrak{g}$. Recall that gauge fields with higher derivative gauge transformations are called "partially-massless" due to the fact that they propagate an intermediary number of degrees of freedom between massless and massive fields [150–152]. The number of derivatives appearing in their gauge transformations is usually called the *depth* and denoted $t$, and hence the massless case corresponds to $t = 1$. Partially-massless fields of arbitrary mixed-symmetry have been analysed thoroughly in [51–53, 60–62], and the particular class considered here, with rectangular Young diagrams, have been discussed in relations with "higher-order" higher spin singletons [153].

Consider the action

$$S[\Psi,\omega] = \int_M \Psi^{A(2s-t+1)}{}_{B(t-1)} H_{AA} \wedge \nabla\omega_{A(2s-t-1)}{}^{B(t-1)}, \qquad 1 \le t \le s, \qquad \text{(A.1)}$$

where, as previously, $\Psi$ and $\omega$ are a 0-form and a 2-form respectively, subject to the gauge transformations

$$\begin{aligned}
\delta_{\xi,\eta}\omega_{A(2s-t-1)}{}^{B(t-1)} &= \nabla\xi_{A(2s-t-1)}{}^{B(t-1)} + e_{A,C} \wedge \eta_{A(2s-t-2)}{}^{B(t-1)C} \\
&\quad + e^{B,C} \wedge \tilde{\eta}_{A(2s-t-1),C}{}^{B(t-2)} - \tfrac{2s-t-1}{2s} \delta_A^B e^{C,D} \wedge \tilde{\eta}_{A(2s-t-2)C,D}{}^{B(t-2)},
\end{aligned} \qquad \text{(A.2)}$$

with $\xi$, $\eta$ and $\tilde{\eta}$ being 1-forms, under which the action is invariant (again thanks to the identities (15) and (16) on the products of vielbeins). These gauge symmetries are also reducible, as it can be verified that the following specific choice of gauge parameters

$$\overset{\circ}{\xi}_{A(2s-t-1)}{}^{B(t-1)} = t\,\nabla\zeta_{A(2s-t-1)}{}^{B(t-1)}, \qquad \text{(A.3a)}$$

$$\overset{\circ}{\eta}_{A(2s-t-2)}{}^{B(t)} = 2t\,(s-t)\,e^{B,C}\zeta_{A(2s-t-2)C}{}^{B(t-1)}, \qquad \text{(A.3b)}$$

$$\overset{\circ}{\tilde{\eta}}_{A(2s-t-1),C}{}^{B(t-2)} = 2s\,(t-1)\left(e_{C,D}\,\zeta_{A(2s-t-1)}{}^{B(t-2)D} - e_{A,D}\,\zeta_{A(2s-t-2)C}{}^{B(t-2)D}\right), \qquad \text{(A.3c)}$$

leads to trivial gauge transformations. The pair $(\Psi,\omega)$ describes a partially-massless field with the symmetry a two-row rectangular diagram, as in the bulk of the paper, but whose curvature has the symmetry

$$\begin{array}{c}
\boxed{\phantom{xx} s \phantom{xx}} \\[-2pt]
\boxed{\phantom{xxxxxx}} \\[-2pt]
\boxed{s-t+1}
\end{array}_{+} \cong \begin{array}{c}
\boxed{t-1}\boxed{\phantom{x}2s-t+1\phantom{x}}
\end{array}, \qquad \text{(A.4)}$$

i.e. it is obtained by taking $s-t+1$ derivatives of the gauge field. The massless case discussed previously corresponds to $t=1$, while for $t>1$ the curvature is defined with *less* derivatives of the gauge field, which reflects the fact that the latter is subject to gauge transformations with more derivatives of the gauge parameter.

As in the massless discussed in Section 2.2, we can package the partially-massless fields of any spin and depth into a generating 2-form,

$$\omega = \sum_{1 \le t \le s} \tfrac{1}{(2s-t-1)!\,(t-1)!} \, y^{A(2s-t-1)} \, \bar{y}_{B(t-1)} \, \omega_{A(2s-t-1)}{}^{B(t-1)}, \qquad \text{(A.5)}$$

and similarly for the 0-forms $\Psi$, as well as the 1-forms $\mathcal{A}$ and the 4-forms $\mathcal{B}$ all of which are assumed to $\mathfrak{g}$-valued. Extending the definition of the pairing (63) to

$$\begin{aligned}
p &: \mathbb{C}[y,\bar{y}] \otimes \mathbb{C}[y,\bar{y}] \longrightarrow \mathbb{C}, \\
f(y,\bar{y}) &\otimes g(y,\bar{y}) \longmapsto \sum_{m,n \in \mathbb{N}} \tfrac{1}{m!n!} \, f_{A(m)}{}^{B(n)} \, g_{B(n)}{}^{A(m)},
\end{aligned} \qquad \text{(A.6)}$$

and the action (78) to all polynomials in $y$ and $\bar{y}$ via

$$p\big(\psi, f \cdot g\big) = p\big(\psi \bullet f, g\big), \qquad \forall\,\psi,f,g \in \mathbb{C}[y,\bar{y}], \qquad \text{(A.7)}$$

one can write the action

$$S[\Psi,\omega;\mathcal{A},\mathcal{B}] = \int_M p \circ \big\langle \Psi, H \wedge \mathcal{D}\omega \big\rangle + p \circ \big\langle \mathcal{B}, \mathcal{F} \big\rangle, \qquad \text{(A.8)}$$

and verify that it is invariant under the gauge transformations

$$\delta_\epsilon \Psi = [\Psi \overset{\bullet}{,} \epsilon]_{\mathfrak{g}}, \qquad \delta_{\xi,\epsilon}\omega = \mathcal{D}\xi + [\omega,\epsilon]_{\mathfrak{g}} + \sigma_+\eta + \tilde{\sigma}_+\tilde{\eta}, \qquad \text{(A.9a)}$$

$$\delta_\epsilon \mathcal{A} = \mathcal{D}\epsilon, \qquad \delta_{\xi,\epsilon}\mathcal{B} = [\Psi \overset{\bullet}{,} \xi \wedge H]_{\mathfrak{g}} + [\mathcal{B} \overset{\bullet}{,} \epsilon]_{\mathfrak{g}}, \qquad \text{(A.9b)}$$

where again, all gauge parameters are understood to be generating functions, $\sigma_+$ is as defined in (48), and $\tilde{\sigma}_+$ is defined implicitly from the part of the free gauge transformations involving $\tilde{\eta}$.

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
