# Peer review of "Massless chiral fields in six dimensions"

_SciPost Physics, doi:SciPost Phys. 19, 079 (2025)_

## Round 1 · Author Response

I am really grateful to all Referees for their careful reading and insightful suggestions, which have contributed to improving my understanding of the subject as well as the quality of this manuscript.

---

## Round 1 · List of Changes

Following the order of the requests in the corresponding reports, here is the list of changes made to the manuscript.

Report #1:
[*] The sentence before 1.21 was slightly modified to clarify that the conformal fields discussed here are the higher spin singletons (later described by Phi and Psi), and footnote 5 was expanded.
[*] I added a sentence below 1.22 to explain that the wavy equal sign is meant to denote equations of motions, I.e. to stress that these are not identities.
[*] A paragraph has been added at the beginning of the subsection "Reducibility of the gauge symmetry" to mention that the number of degrees of freedom per field is 2s+1, and containing a reference to the papers suggested by the Referee.
[*] I changed diagram —> (sub)diagram below 1.42.
[*] This is indeed a misleading abuse of terminology. Accordingly, the term ‘Yang—Mills field’ has been replaced by 1-form connection or spin-1 field throughout the paper as suggested.
[*] Corrected ‘verifies' with ‘satisfies' before 1.55.
[*] The coupling constant has been removed from the paper since, as rightly pointed out by the Referee, it can be absorbed by a field redefinition, and is not used in any meaningful way.
[*] The sentences around eq. 1.60 and 1.61 have been slightly re-written, please check whether clarity has been improved sufficiently.
[*] I am grateful to the referee for clarifying what was a point of confusion for me. Consequently, the F-dependent terms in the gauge transformations were removed throughout the paper, as the expressions become simpler, and footnote 8 was added merely to point out that the gauge transformations can be made reducible off-shell by modifying them with a term linear in F. The last paragraph of Section 1.1 was also removed, as it was meant to introduce \sigma_-^\dagger, which is not useful anymore.
[*] The sentence between 1.73 and 1.74 (now eq. 1.71 and 1.72) was split into two, hopefully improving readability.
[*] Removed ‘that’ after 1.76.
[*] Footnote 9 has been added to stress that the cubic term 1.85 is a specificity 6d chiral theory.
[*] A sentence has been added below 1.95 to refer to Ref. [3] of the report (Ref. [132] of the revised manuscript).

Report #2:
[*] The mention of sl(2,H)-tensors has been replaced throughout the paper with su^*(4)-tensors.
[*] There was a typo here, which has been corrected: the relevant equation is H \wedge \nabla A = 0 instead of H \wedge A = 0. The purpose of this short paragraph was simply to describe what kind of fields could be embedded in A. As the referee point out, they form a topological background, their equation of motion being F=0. However, if the 1-form A was subject to another kind of equation of motion, it may describe propagating degrees of freedom. An example of such an equation is H \wedge \nabla A, whose solutions do propagate degrees of freedom corresponding to massless fields of symmetry 1.71.
[*] Indeed, the coupling to a BF-theory seems compatible with the introduction of nonlinearities in the zero-form, provided one finds an equivariant map as argued in the last paragraph of section 1, added in the revised version. The existence of such an object will of course depend on the type of Lie algebra \mathfrak{g} and is however not garanteed (at least to the best of my knowledge).
[*] Current interactions are in principle also available, and the interesting feature of models involving a 2-form gauge field is that they naturally allow for the simplest type of higher-form symmetry, namely 1-form symmetries. A couple of short paragraph containing this comment have been added at the end of Section 1.2.

Report #3 & #4:
The notation for the basis of forms has been changed as follows: the 2-forms previously denoted by H_A{}^B, which was indeed a source of confusion, are now denoted by \Sigma_A{}^B, while the 4-forms \hat H_A{}^B have been simply replaced by a product of the aforementioned 2-forms (since their use in the paper was in fact quite limited, it seems it does not deserve the introduction of a new symbol).

---

## Editorial Decision

published